# Mitochondrial movement during its association with chloroplasts in *Arabidopsis thaliana*

Kazusato Oikawa[1], Takuto Imai[2], Chonprakun Thagun[2], Kiminori Toyooka [3], Takeshi Yoshizumi[2], Kazuya Ishikawa[4], Yutaka Kodama [2,4✉] & Keiji Numata [1,2✉]

Plant mitochondria move dynamically inside cells and this movement is classified into two types: directional movement, in which mitochondria travel long distances, and wiggling, in which mitochondria travel short distances. However, the underlying mechanisms and roles of both types of mitochondrial movement, especially wiggling, remain to be determined. Here, we used confocal laser-scanning microscopy to quantitatively characterize mitochondrial movement (rate and trajectory) in *Arabidopsis thaliana* mesophyll cells. Directional movement leading to long-distance migration occurred at high speed with a low angle-change rate, whereas wiggling leading to short-distance migration occurred at low speed with a high angle-change rate. The mean square displacement (MSD) analysis could separate these two movements. Directional movement was dependent on filamentous actin (F-actin), whereas mitochondrial wiggling was not, but slightly influenced by F-actin. In mesophyll cells, mitochondria could migrate by wiggling, and most of these mitochondria associated with chloroplasts. Thus, mitochondria migrate via F-actin-independent wiggling under the influence of F-actin during their association with chloroplasts in *Arabidopsis*.

[1] Department of Material Chemistry, Graduate School of Engineering, Kyoto University, Kyoto, Japan. [2] Biomacromolecules Research Team, RIKEN Center for Sustainable Resource Science, Saitama, Japan. [3] Mass Spectrometry and Microscopy Unit, RIKEN Center for Sustainable Resource Science, Yokohama, Japan. [4] Center for Bioscience Research and Education, Utsunomiya University, Utsunomiya, Japan. ✉email: kodama@cc.utsunomiya-u.ac.jp; keiji.numata@riken.jp

Mitochondria are essential organelles involved in many cellular processes, such as adenosine 5′-triphosphate energy production, $Ca^{2+}$ homeostasis, and the regulation of reactive oxygen species generation[1–3]. In some metabolic pathways, mitochondria function in cooperation with chloroplasts. For example, oxidative electron transport, phosphorylation, and photorespiration in mitochondria are essential for sustaining photosynthesis in chloroplasts[4,5]. Examination of static micrographs suggests that mitochondria physically associate with chloroplasts to facilitate the efficient exchange of metabolites between the two organelles[4,6]. Other studies have reported that mitochondria colocalize with chloroplasts by transmission electron microscopy and epifluorescent microscopy[7–10].

In plant cells, mitochondria directionally move at various speeds depending on the cell type, tissue, and plant species[11–15]. The maximum speed of this directional movement is 2.5–10 μm s$^{-1}$ in elongated tobacco (*Nicotiana tabacum*) culture cells, *Picea wilsonii* pollen tubes, and *Arabidopsis thaliana* root hair cells[11–13]. Filamentous actin (F-actin) mediates the directional movement of mitochondria, and microtubules participate in mitochondrial positioning by controlling the orientation of F-actin[11–13]. In many pharmacological studies, F-actin-disrupting drugs such as latrunculin B, cytochalasin D, and jasplakinolide (in conjunction with the myosin inhibitor 2,3-butanedione 2-monoxime) arrested the directional movement of mitochondria[11–13]. During treatment with F-actin-disrupting drugs, F-actin-independent wiggling of mitochondria was observed[12,13]. This wiggling has also been referred to as "Brownian movement" or "Brownian motion" in previous reports[14,15]. Although mitochondrial wiggling has been observed in many previous studies, the characteristics of this wiggling remain to be described in detail. Moreover, mitochondria lose their motility under long-term illumination with weak blue light at levels that activate chloroplast metabolism, suggesting that mitochondrial motility is reduced through their association with chloroplasts whose metabolism is activated[16]. This report suggested that mitochondrial movements would be influenced by chloroplast[16]. However, to date, how the mitochondrial movement is influenced by its association with chloroplast has not been clarified.

Here, we quantitatively characterized two types of mitochondrial movement, directional movement and wiggling of mitochondria, using time-lapse video recording of over 1000 mitochondria in transgenic *A. thaliana* protoplasts and leaf palisade mesophyll cells. Quantitative time-lapse analysis with and without inhibitors indicated that mitochondria migrate via F-actin-independent wiggling during their association with chloroplasts in *A. thaliana*.

## Results

### Two distinct types of mitochondrial movement in *Arabidopsis*.
To observe mitochondria in living plant cells, we produced transgenic *A. thaliana* expressing Citrine yellow fluorescent protein fused with a mitochondrial targeting sequence (MTS-Citrine). To evaluate mitochondrial motility in single plant cells, we examined protoplasts isolated from the mesophyll cells of transgenic plant leaves. In the transgenic protoplasts, Citrine-fluorescent mitochondria were clearly observed via confocal laser-scanning microscopy (CLSM; Fig. 1a). When the cells were observed at 5-s intervals, two distinct types of mitochondria were recognized: some mitochondria directionally moved to different locations, while the others did not change their locations (Fig. 1b). Time-lapse video recording every 250 ms confirmed the two distinct types of mitochondrial movement, namely, directional movement (in which mitochondria travel long distances) and wiggling (in which mitochondria move but remain in almost

the same position on chloroplasts; Supplementary Movie 1). To quantify the speeds of the directional movement and wiggling, we tracked more than 100 mitochondria in time-lapse images acquired every 250 ms for 30 s (Fig. 1c). Frequency analysis indicated that the mitochondria moved at various speeds from 0.05 to 1.00 μm s$^{-1}$ (Fig. 1d).

### Characterization of mitochondrial movements.
To further characterize the directional movement and wiggling of mitochondria, we focused on the migration distance, speed, and angle changes of the mitochondrial movements (Fig. 2). When we measured the straight-line distance between the first and last positions of mitochondria in the trajectories within 30 s (Fig. 2a), the frequency of the distance gradually decreased until 5 μm, and the subsequent peaks appeared at longer distances (>5 μm; Fig. 2b). Similar results were obtained in leaf palisade mesophyll cells (Supplementary Fig. 1a, b). Based on these results, we separated the mitochondria into two groups: the short-distance (<5 μm) and long-distance (5 μm <) groups.

In addition, we measured and characterized the speed and angle changes of the mitochondrial movements at every 1 s time point for 30 s. Scatter plot of the speed and angle changes showed two distinct regions: low speed with high angle changes (gray region) and high speed with low angle changes (pale-blue region; Fig. 2c). Trajectories and scatter plot for the speed and angle changes of three representative mitochondrial movements from each region (gray and pale-blue regions in Fig. 2c) are shown in Fig. 3: mitochondria from the gray region showed short-distance migration (i.e., wiggling, distance <5 μm; Fig. 3a upper) at low speed with high angle changes (Fig. 3b upper), and mitochondria from the pale-blue region showed long-distance migration (i.e., directional movement, 5 μm < distance; Fig. 3a lower) at high speed and low angle changes (Fig. 3b lower). We detected a similar trend in leaf palisade cells (Supplementary Fig. 1c). These different types of mitochondria, which were separated based on migrate distance, had statistically different mean speeds and angle changes (Supplementary Fig. 2). However, a long-distance migration of mitochondria contains a few of the plot of low speed and high angle changes (Fig. 3b, 5 μm < distance).

Taken together, mitochondria migrating <5 μm in 30 s moved largely at low speed with high-angle change rates defined as wiggling, and mitochondria migrating >5 μm in 30 s moved largely at high speed with low-angle change rates defined as directional movement. These results indicate that our method for evaluating and quantifying mitochondrial movements was sufficient to further explore the differences between wiggling and directional movement.

### Mitochondrial wiggling is independent of F-actin.
To quantitatively examine the contribution of the cytoskeleton (i.e., F-actin and microtubules) to directional movement and wiggling, we treated transgenic protoplasts with the F-actin-disrupting drug cytochalasin B (cytochalasin) and the microtubule-disrupting drug oryzalin[17,18] and analyzed the mitochondrial migration distance, speed, and angle changes (Fig. 4, Supplementary Figs. 3 and 4, and Supplementary Movies 1–6). Scatter plot derived from each trajectory of mitochondria showed that mitochondria in cytochalasin-treated cells, directional movement did not occur (Fig. 4a). Instead, almost all mitochondria showed short-distance migration at low speed and high angle changes on or beside chloroplasts (Fig. 4b, c and Supplementary Movie 7), which was similar to the behavior of wiggling mitochondria (gray area in Fig. 2c), however the speed was suppressed compared to normal cell (Supplementary Fig. 4). By contrast, mitochondria in oryzalin-treated cells showed both directional movement and

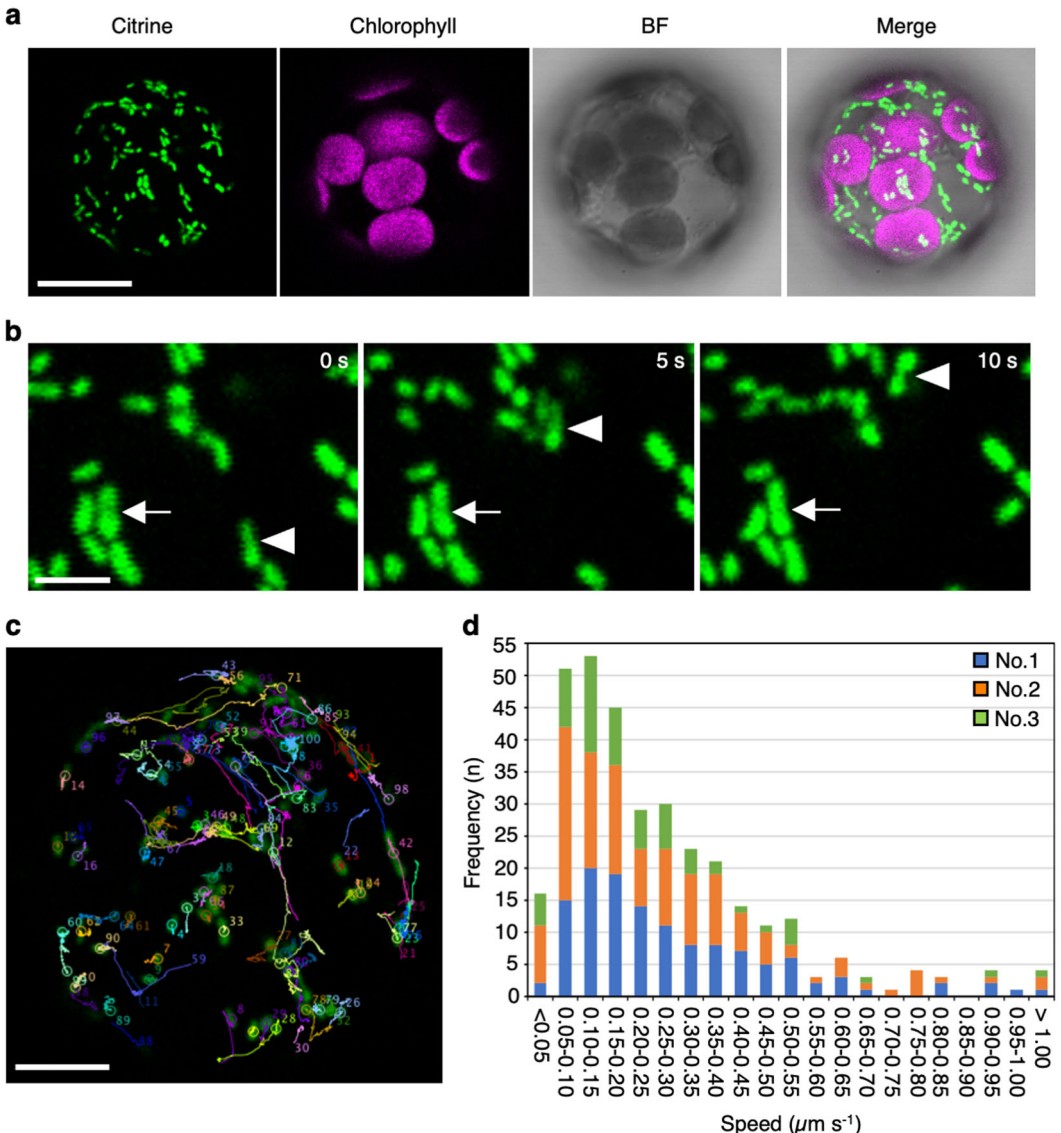

**Fig. 1 Observation of mitochondrial movement in mesophyll protoplasts. a** Citrine fluorescence in mitochondria (green) and chlorophyll autofluorescence (magenta), and bright field (BF) image of mesophyll protoplasts of MTS-Citrine transgenic *A. thaliana* are shown as a merged image (Merge) taken by CLSM. Scale bar: 10 μm. **b** Time-lapse images of mitochondria taken every 5 s. Fast, directional movements (arrowheads) and slow, wiggling movements (arrows) of mitochondria are shown. Scale bar: 2 μm. **c** Representative images showing the trajectories of mitochondrial movements over a 30-s period. Mitochondrial movements were tracked and quantified using Fiji software and its plugin MTrackJ. Mitochondria are labeled with numbers and outlined with different colored lines. Scale bar: 10 μm. **d** Speed distribution of mitochondrial movement from three independent protoplasts (protoplast No. 1, blue; protoplast No. 2, orange; and protoplast No. 3, green).

wiggling, as did mitochondria in DMSO-treated cells (Fig. 4a, c). However, plots of speed (Fig. 4c) and speed frequency (Supplementary Fig. 4) at >0.5 μm s$^{-1}$ were slightly decreased in oryzalin-treated cells. Therefore, both directional movement and wiggling occurred independently of microtubules, but microtubule may have effect on mitochondria directional movement. These results indicate that directional movement depends on F-actin, whereas wiggling occurs independently of F-actin and microtubules.

To confirm the observation that wiggling occurs independently of F-actin, we simultaneously observed both F-actin and mitochondria in *A. thaliana* cells. We transiently expressed Lifeact-Citrine, which visualize F-actin[19], in protoplasts isolated from transgenic *A. thaliana* harboring mitochondria with red fluorescent protein (RFP) fused to the F1-ATPase delta-prime subunit[20] (Fig. 5). When mitochondria and F-actin on chloroplast surface were observed, we found that mitochondria in close proximity to F-actin on chloroplast (Supplementary Fig. 5 and Supplementary Table 1). When most F-actin was disrupted in protoplasts treated with 50 μM cytochalasin (the Citrine showed diffuse localization in the cytosol; Fig. 5a), number of mitochondria in close proximity to F-actin was reduced on chloroplast (Fig. 5b). However, mitochondria remained to associate with chloroplasts without F-actin (Fig. 5c and Supplementary Movie 8). Even when we increased the cytochalasin concentration to 500 μM, which exceeds the concentration of cytochalasin strongly inhibit F-actin polymerization[17,18], mitochondria still associated with chloroplasts (Fig. 5b, c and Supplementary Movie 8). In 500 μM cytochalasin-treated protoplasts, the mitochondrial movement was confirmed to be the wiggling by analyzing the tracking and speed – angle changes of mitochondrial movement (Supplementary Fig. 6), which shows similar pattern to the

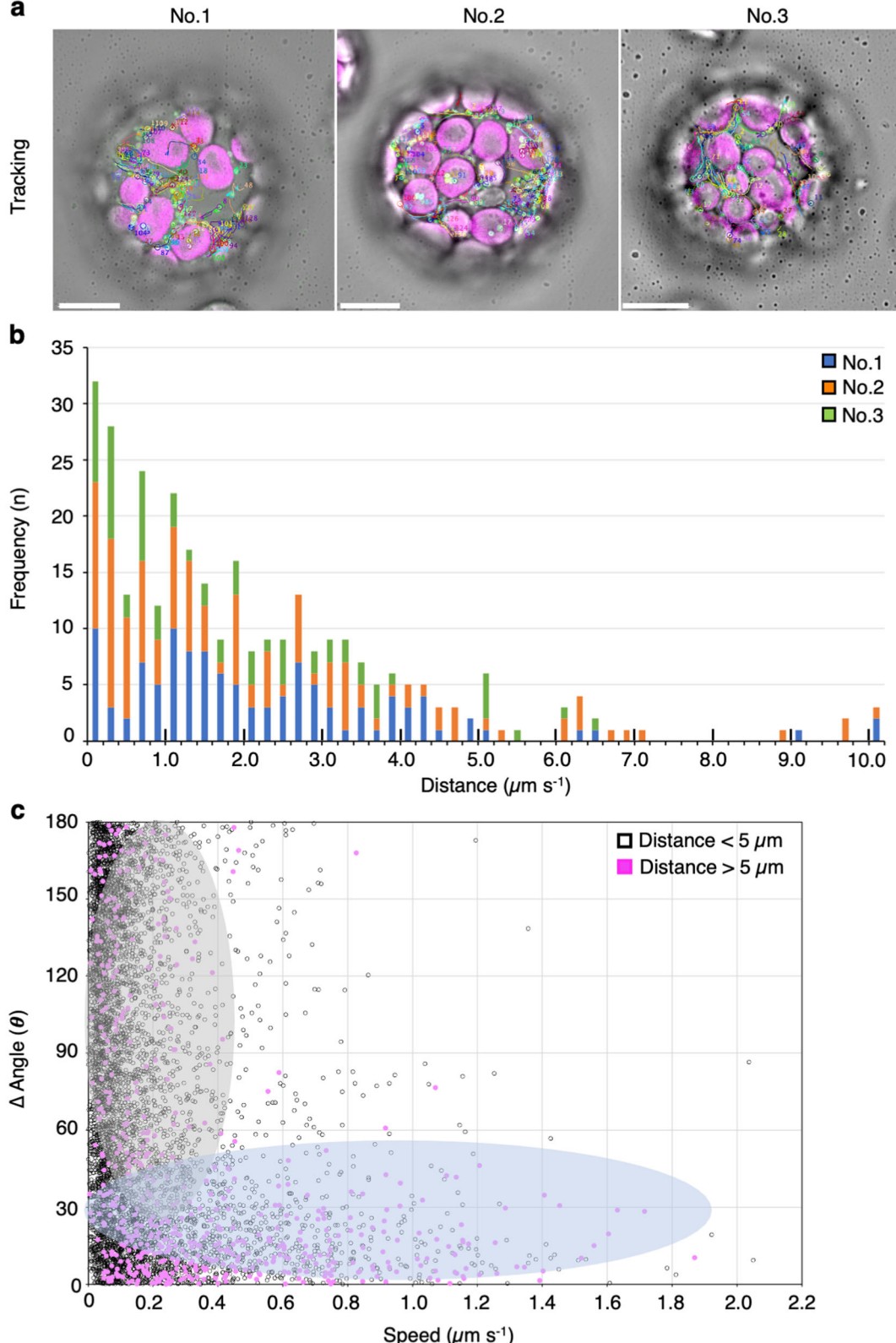

**Fig. 2 Characterization of mitochondrial movements in leaf mesophyll protoplasts. a** Trajectories of mitochondrial movements constructed from time-lapse analysis of images of three different protoplasts (No. 1–3) acquired for 30 s at 250-ms intervals. Scale bar; 10 μm. **b** Distribution of the direct distance between the first and last point of mitochondria in the trajectories of mitochondrial movement acquired from three independent protoplasts (protoplast No. 1, blue color; protoplast No. 2, orange; and protoplast No. 3, green). **c** Scatter plot of speed (x-axis) and angle changes (y-axis) of mitochondria at each time point acquired from the trajectories of mitochondrial movement, which are separated based on a distance shorter (open circles) or longer (filled-magenta circles) than 5.0 μm in **b**. The plots are separated into two regions highlighted in gray and pale blue.

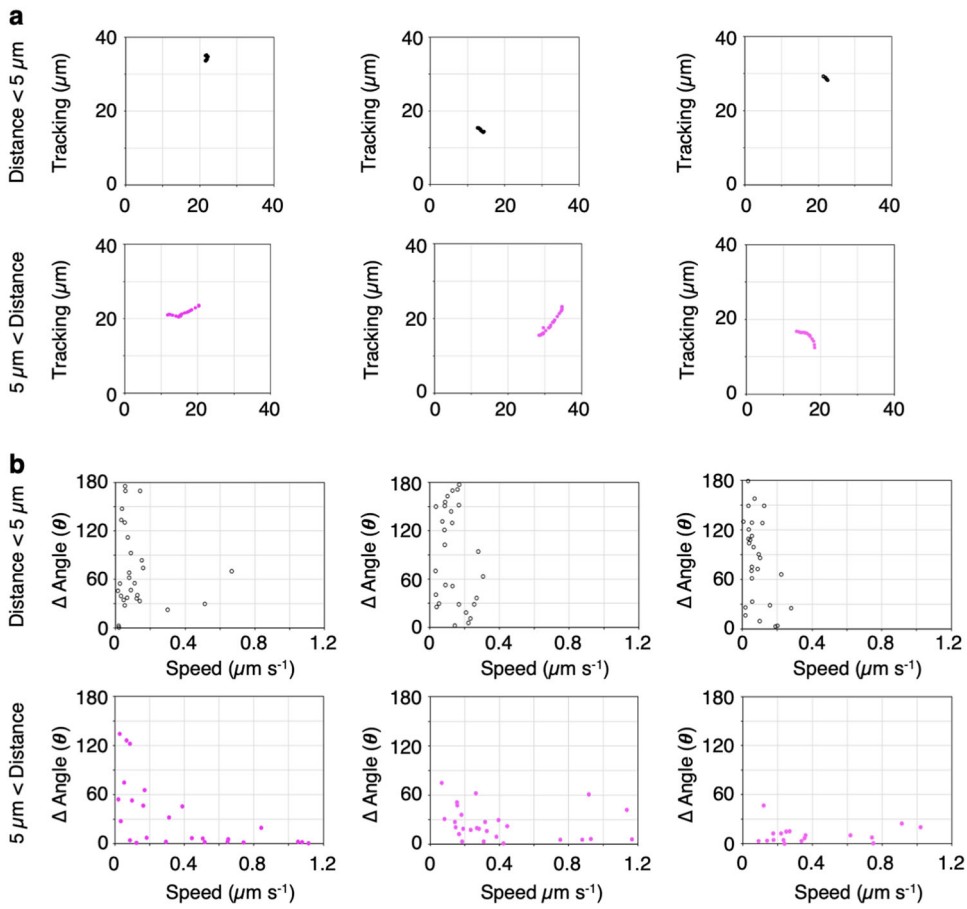

**Fig. 3 Different patterns of movement between mitochondria migrating at a shorter or longer distance than 5.0 μm. a, b** Representative trajectories (**a**) and scatter plot of changes in speed and angle (**b**) of three mitochondrial movements with shorter (upper panel) or longer (lower panel) than 5.0 μm migration distance in Fig. 2.

wiggling in Figs. 2–4. About 81% of mitochondria revealed the wiggling on chloroplast in 500 μM cytochalasin-treated protoplasts (Supplementary Fig. 7). These results confirm that mitochondrial wiggling on chloroplasts occurs independently of F-actin.

Frequency analysis of the mean speeds of mitochondria indicated that the higher speed fraction ($0.35\ \mu m\ s^{-1} <$) disappeared in cytochalasin-treated cells and the lower speed fraction ($<0.35\ \mu m\ s^{-1}$) greatly increased (Supplementary Fig. 4). The results indicate that F-actin-dependent directional movement and F-actin-independent wiggling can be distinguished not only by distance but also by speed.

**Mitochondria can migrate via wiggling.** To clarify whether mitochondria can migrate via F-actin-independent wiggling, we generated videos using time-lapse images of cytochalasin-treated cells and chemically fixed cells (Fig. 6a) at 250-ms and 5-s intervals. In the cytochalasin-treated cells, the mean distance per 5-s interval of mitochondria was $0.61 \pm 0.35$ μm, which was ~2-fold greater than that per 250-ms interval ($0.35 \pm 0.14$ μm; Fig. 6b), suggesting that mitochondria can migrate in the intracellular space via wiggling. However, time-lapse fluorescence images can show false mitochondrial movement due to random fluctuations in fluorescence, a type of technical noise.

To confirm that the observed wiggling was distinct from technical noise, we estimated false mitochondrial movement using chemically fixed cells in which mitochondrial movement was completely lost (Supplementary Fig. 4 and Supplementary

Movie 9). In the fixed cells, no difference was observed between the mean migration distance at the 250-ms ($0.33 \pm 0.08$ μm) and 5-s time intervals ($0.30 \pm 0.13$ μm; Fig. 6c). Therefore, by changing the time interval in the time-lapse video, mitochondrial wiggling could be distinguished from fluorescence fluctuations. Indeed, scatter plot of changes in speed vs. angle changes showed that mitochondria in fixed cells had lower speeds and higher angle changes than those in cytochalasin-treated cells (Fig. 6d, e). Overall, these findings indicate that mitochondria can migrate via F-actin-independent wiggling in *A. thaliana* mesophyll cells.

**Wiggling mitochondria associate with chloroplasts.** Consistent with previous reports[7,10], we detected the colocalization of mitochondria with chloroplasts (Fig. 1a and Supplementary Movie 10). This colocalization was also observed in cytochalasin- and oryzalin-treated cells (Figs. 4b and 5b and Supplementary Movies 3–8), indicating that the association of the two organelles is independent of F-actin and microtubule polymerization. To further explore the relationship between mitochondrial movement and their association with chloroplasts, we counted the number of mitochondria that did or did not associate with chloroplasts over a 30-s period in time-lapse images (Fig. 7a, b and Supplementary Table 2). We classified these different types of mitochondria into three groups based on their interactions with chloroplasts in a time-dependent manner: (1) the no association group, NA; (2) the partial association group, PA (more than 3 s); and (3) the continuous association group, CA (within 30 s). Approximately 17% of the mitochondria were not associated with

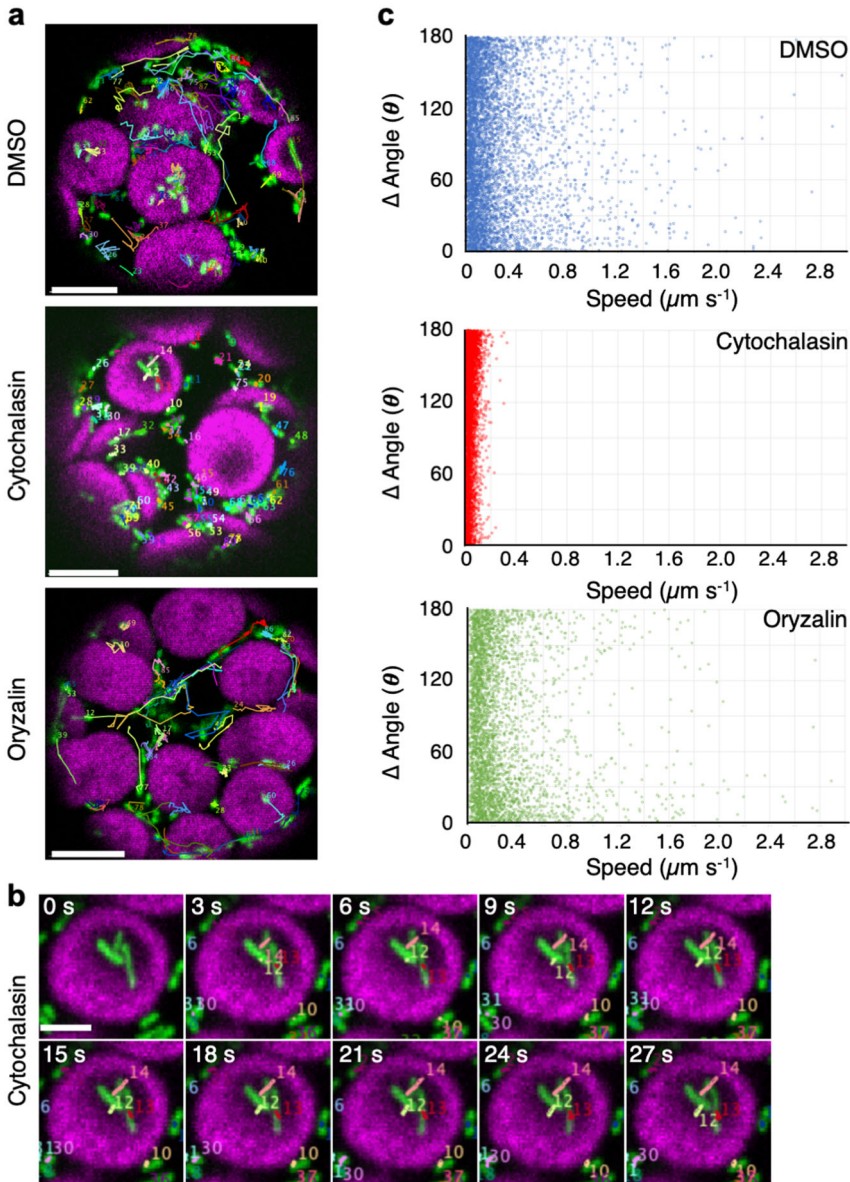

**Fig. 4 Characterization of mitochondrial movement in the presence of cytoskeletal inhibitors. a** Trajectories of mitochondrial movements in DMSO-, cytochalasin B (cytochalasin)-, and oryzalin-treated protoplasts acquired from time-lapse analysis of images taken for 30 s at 250-ms intervals. Scale bar; 5 μm. **b** Time-lapse images of mitochondria on the chloroplast surface in cytochalasin-treated protoplasts at 3-s intervals. Scale bar: 2 μm. **c** Scatter plot of speed (x-axis) and angle changes (y-axis) of mitochondria at each time point acquired from the trajectories of mitochondrial movements in **a**. Protoplasts treated with DMSO (blue open circles), cytochalasin (red open circles), and oryzalin (green open circles).

chloroplasts, ~41% of the mitochondria partially associated with chloroplasts, and ~42% of the mitochondria continuously associated with chloroplasts (Fig. 7b).

We measured the speed and angle changes of mitochondrial movements at each time point and summarized the information in scatter plot to distinguish among the NA, PA, and CA groups (Fig. 7c). Data in the scatter plot were separated into two distinct regions: mitochondria that moved at high speed with low angle changes (NA and PA), and mitochondria that moved at low speed with high angle changes (CA) (Fig. 7c). Three representative trajectories and scatter plot for speed vs. angle changes of single mitochondria were constructed for the three groups (Fig. 8). Both NA and PA mitochondria showed long-distance directional movement (Fig. 8a, NA, PA) at high speed with low angle changes (Fig. 8b, NA, PA). By contrast, CA mitochondria showed wiggling and short-distance migration (Fig. 8a, CA) at low speed

with high angle changes (Fig. 8b, CA). Similar results were obtained in leaf palisade cells (Supplementary Figs. 8 and 9 and Supplementary Table 3), in which wiggling specifically occurred adjacent to chloroplasts (Supplementary Movie 11).

We performed frequency analysis of the mitochondrial speeds in each group (Supplementary Fig. 10). The speed frequency of mitochondria in the NA group was >0.40 μm s⁻¹ (Supplementary Fig. 10), suggesting that directional movement occurred in the cytosol. The speed frequency of mitochondria in the PA group ranged from 0.10 to 0.55 μm s⁻¹ (Supplementary Fig. 10), suggesting that they frequently moved between chloroplasts and the cytosol. The speed frequency of mitochondria in the CA group peaked at 0.05–0.10 μm s⁻¹ (Supplementary Fig. 10), a speed similar to that of wiggling mitochondria in cytochalasin-treated cells (Supplementary Fig. 4). These results indicate that mitochondrial speeds are correlated with interactions with

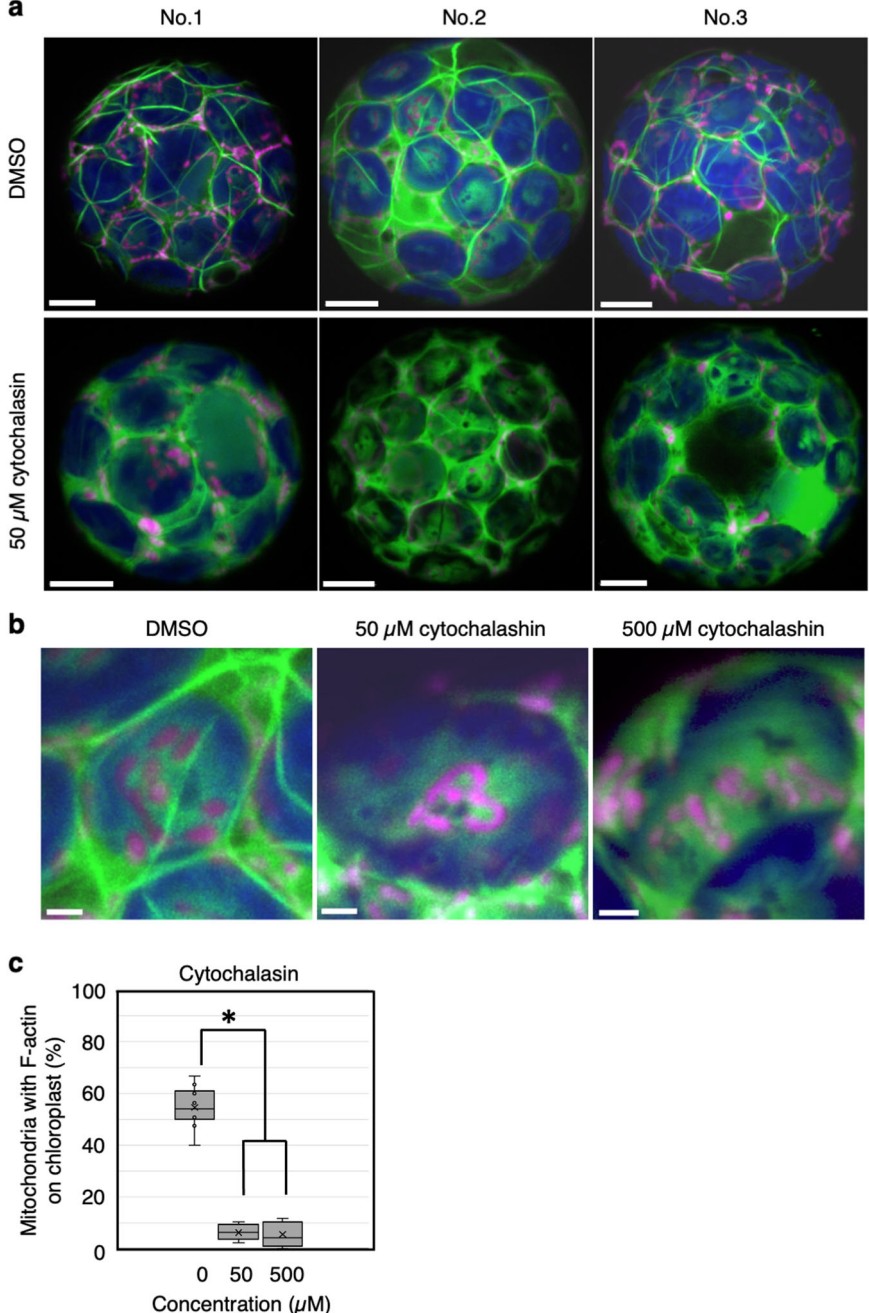

**Fig. 5 F-actin-independent association between mitochondria and chloroplasts. a** Representative images of F-actin (green) and mitochondria (magenta) in three different protoplasts treated without (DMSO) or with 50 μM cytochalasin B (cytochalasin). Scale bars: 5 μm. **b** Enlarged images of mitochondria and F-actin on the chloroplast surface in protoplasts treated with 0 μM, 50 μM, and 500 μM cytochalasin. Scale bars: 5 μm. **c** Quantification of the number of mitochondria associated with F-actin on chloroplasts. The number of mitochondria in protoplasts without (DMSO) vs. with 50 μM or 500 μM cytochalasin differed (Student's *t* test, *<0.01).

chloroplasts. Mitochondrial movement slowed in the absence of F-actin, and they started wiggling at lower speeds on the chloroplast surface. Taken together, these findings indicate that mitochondrial wiggling is induced by interactions with chloroplasts in *A. thaliana*.

**MSD analysis of mitochondrial movement**. To confirm characteristics of mitochondrial movements, we performed mean-squared displacement (MSD) analysis[21–24] (Eq. 1; see MSD calculation in Methods section) on the trajectories of each

mitochondrial movement in Figs. 2c, 4c, 6d, e, and 7c. The two-dimensional MSD plots revealed linear or parabolic shape, and were fitted to Eqs. 2 or 3 (see MSD calculation in Methods section, Supplementary Figs. 11–13 and Supplementary Table 4). The fixed cell revealed linear shape defined as Brownian motion with low diffusion (*D*; 0.00023)[23–25] (Supplementary Fig. 11f). Mitochondrial movement with long-distance migration (>5 μm), DMSO-treatment, oryzalin-treatment, NA, and PA exhibiting directed diffusion curve with steeper slope, meaning directed motion with high velocity (*v*: 0.406, 0.289, 0.263, 0.254, and 0.267, respectively; Supplementary Figs. 11b, c, d and 12a, b). On the

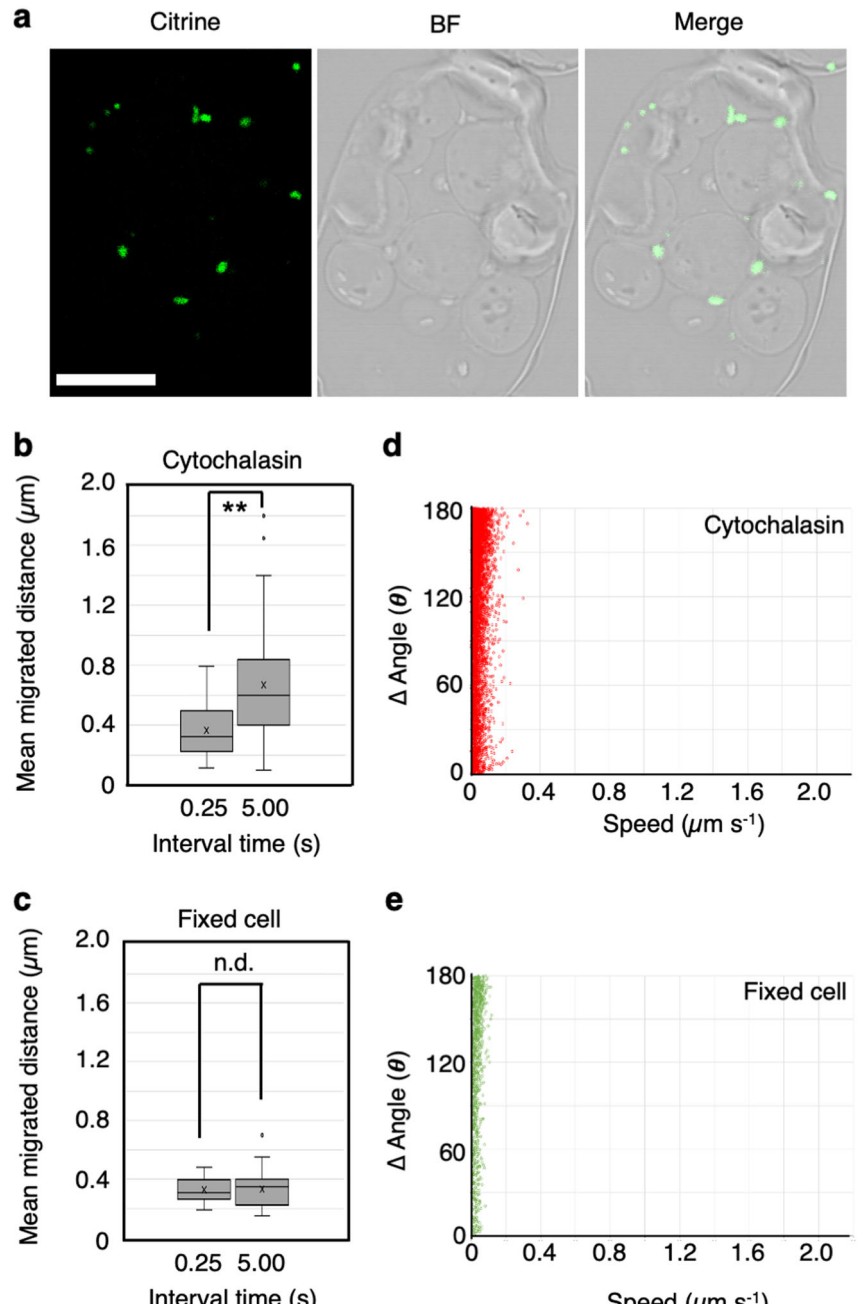

**Fig. 6 Mitochondrial movement in cytochalasin-treated protoplasts is different from that in fixed cells. a** CLSM images of mitochondria (Citrine, green) and bright field (BF) images of fixed cells were merged (Merge). Scale bar: 10 μm. **b, c** Mean migration distance of mitochondria in cytochalasin-treated protoplasts (**b**) and in fixed cells (**c**) from time-lapse videos taken at 0.25-s and 5-s intervals. The values represent the mean ± S.D. in **b** and **c**. **\*\****P* < 0.01, n.d. *P* > 0.05 (Student's *t* test). N.d. no difference. **d, e** Scatter plot of speed (*x*-axis) and angle changes (*y*-axis) of mitochondria at each time point acquired from the trajectories of mitochondrial movements in three fixed protoplasts. Each plot is shown as cytochalasin-treated (**d**) and fixed (green) protoplasts (**e**).

other hand, mitochondrial movement with short-distance migration (<5 μm) and CA revealed both directed- and diffusive motion with slow slope (*D*: 0.11 and 0.029, respectively) (*v*: 0.0332 and 0.0436, respectively; Supplementary Figs. 11a and 12c and Supplementary Table 4). This means that CA shows negligible directed motion compared with NA, and seems to be wiggling. Mitochondrial movement with cytochalasin-treatment also revealed both directed- and diffusive motion with slow slope, but low velocity (*v*: 0.0100) and low diffusion (*D*: 0.00010) (Supplementary Fig. 11e and Supplementary Table 4). These MSD results conclude that the wiggling appears to be similar to diffusive motion with low velocity, and that directional movement has high velocity, supporting the previous conclusion in this study.

## Discussion

Many studies have reported that mitochondria cease their directional movement when treated with F-actin-disrupting drugs[12-15], but they continue to exhibit wiggling in various cell types in several plant species[12-14]. These studies defined the relationship between F-actin and directional movement but did not address wiggling[12-15]. In the present study, we analyzed

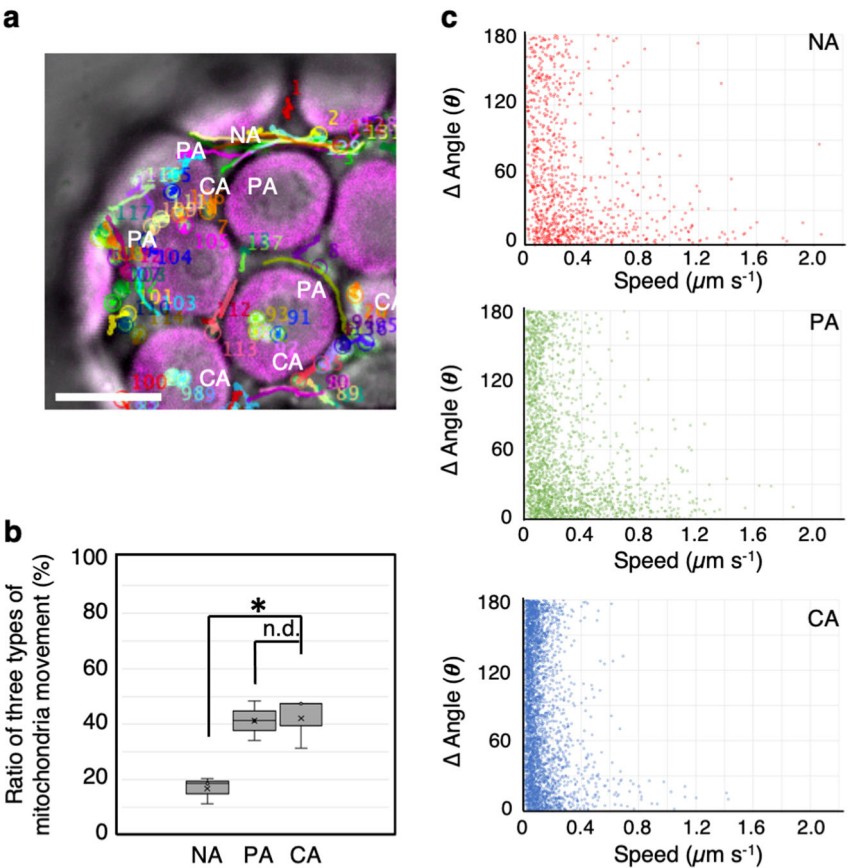

**Fig. 7 Mitochondrial movement is influenced by association with chloroplasts. a** CLSM image representing three types of mitochondrial movements: no association (NA), partial association (PA), and continuous association (CA) with chloroplasts. Trajectories of mitochondrial movement constructed from time-lapse analysis of images acquired at 30-s at 250-ms intervals. Scale bar: 10 μm. **P < 0.01, n.d. P > 0.05 (Student's t test). N.d. no difference. **b** Ratio of mitochondrial movement classified as having NA, PA, and CA with chloroplasts. **c** Scatter plot of speed (x-axis) and angle changes (y-axis) of three different types of mitochondrial movements, including, NA (red), PA (green), and CA (blue) at each time point acquired from the trajectories of mitochondrial movement in three different protoplasts.

F-actin-dependent directional movement as well as F-actin-independent wiggling in leaf mesophyll protoplasts of *A. thaliana*. We first separated the mitochondria into two groups depend on migrate distance as 5 μm border, which the second peak appeared in distribution of the migrate distances. We thought that the 5 μm is related to chloroplast diameter, which mitochondria in CA associate with. Then, we determined that wiggling mitochondria exhibited short-distance migration at low speed and high angle changes and that they associated with chloroplasts.

In addition, we applied our methods for evaluating mitochondrial movement to intact leaf mesophyll cells and obtained similar results (Supplementary Figs. 1, 8, 9, and 14 and Supplementary Movie 11). Our findings confirm the notion that mitochondrial wiggling generally occurs in leaf mesophyll cells. Frequency analysis of the mean speeds of mitochondria revealed that the high-speed fraction was slightly larger in leaf mesophyll cells (Supplementary Fig. 14a) than that in protoplasts (Supplementary Fig. 4a, DMSO). Both the mean and maximum speeds of mitochondrial movement were higher in leaf mesophyll cells than that in protoplasts (Supplementary Fig. 14b, c), likely due to differences in cell shape or culture conditions between protoplasts and intact leaf cells. Since mitochondria in both cell types are active, we mainly used leaf mesophyll protoplasts to obtain clear image with avoiding contamination of mitochondrial images from leaf epidermis cells. Leaf mesophyll protoplasts is useful material for fluorescence imaging in only mesophyll cell.

Consistent with previous studies[12–15], we determined that the directional movement of mitochondria was controlled by F-actin and that the wiggling was induced independently of F-actin (Figs. 4 and 5, Supplementary Figs. 3 and 4, and Supplementary Movies 5–8). Our results suggest that the driving force for the directional movement of mitochondria is F-actin-mediated motility. However, the driving force for mitochondrial wiggling remains to be determined. A previous in vitro study indicated that mitochondria isolated from tobacco pollen tubes move on microtubule rails[26], suggesting that microtubules are a driving force for wiggling. However, treatment with oryzalin, a microtubule-disrupting drug, did not inhibit mitochondrial wiggling, whereas directional movement seemed to be slightly reduced in *A. thaliana* mesophyll protoplasts (Fig. 4a, c, Supplementary Fig. 4, and Supplementary Movies 3, 4), suggesting that this wiggling occurs independently, but the directional movement would be slightly affected by microtubules and related motors, such as kinesins. The result would be related to microtubule function affecting actin filament organization leading to affecting mitochondria velocity and trajectory in directional movement[11,12], or to the event that mitochondria trapped at F-actin - microtubule junction[27]. In the current study, the MSD analysis showed that mitochondrial movement with a short-distance migration (<5 μm) in oryzalin-treated cell maintained both directed- and diffusive motion with slow slope (D: 0.038, v: 0.134; Supplementary Fig. 13 and Supplementary Table 4), which is characteristic of the mitochondrial wiggling. Thus, it

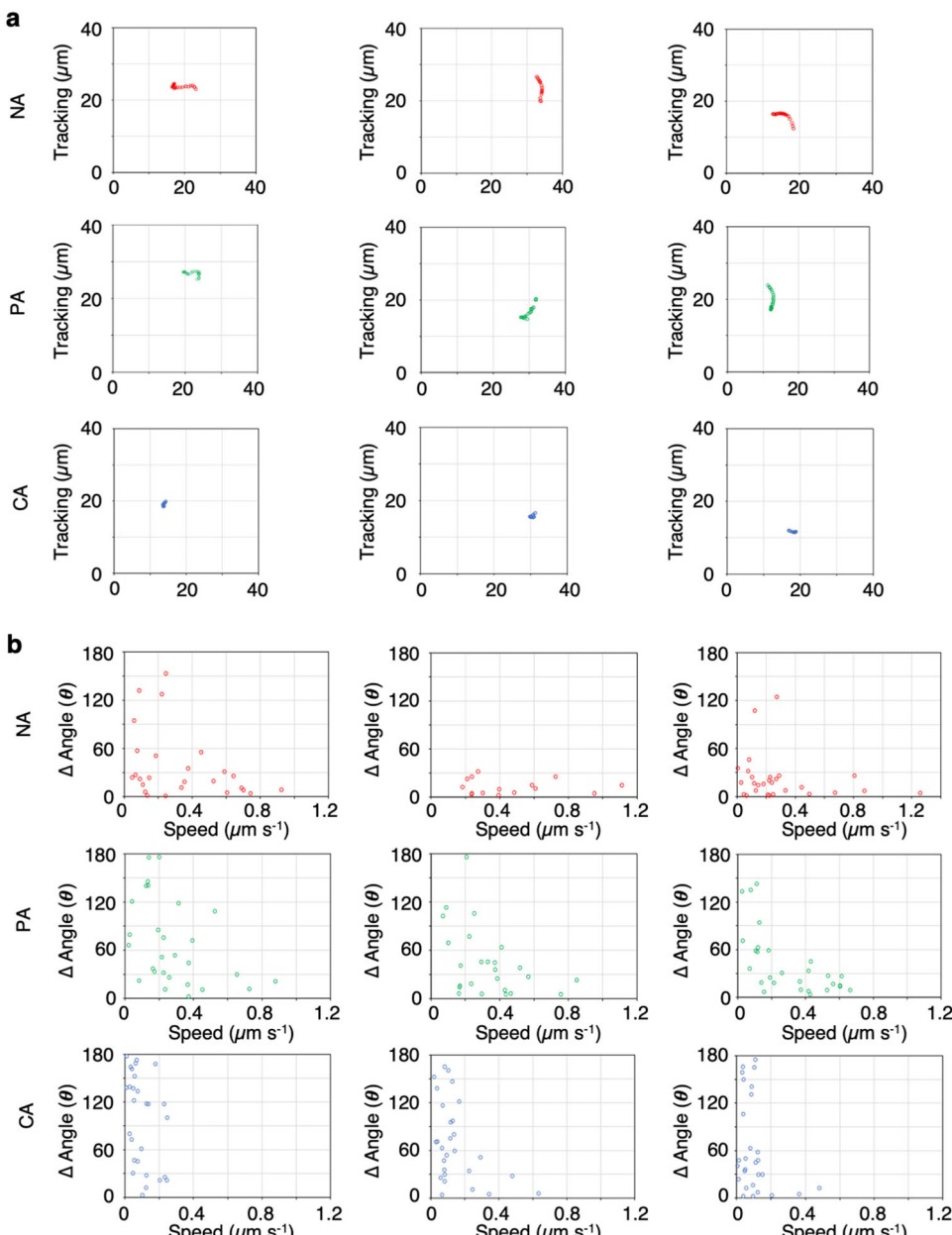

**Fig. 8 Different patterns of mitochondrial movement depend on their association with chloroplasts. a, b** Representative trajectories (**a**) and scatter plot of speed and angle changes (**b**) of three mitochondrial movements were classified as no association, NA (upper panels); partial association, PA (middle panels); and continuous association, CA (lower panels) with chloroplasts.

appears that microtubules are not a driving force for the mito-chondrial wiggling. Further studies were needed to identify the driving force for mitochondria wiggling.

To develop methods for evaluating mitochondrial movements, we focused on mitochondrial behaviors (migration distance, speed, and angle changes) and performed rapid scanning of time-lapse images obtained via CLSM. Using our developed evaluation methods, we distinguished between directional movement and wiggling and further distinguished between wiggling and tech-nical noise (fluorescence fluctuation; Figs. 2, 4, 6, and 7). Using speed-based frequency analysis, we also successfully evaluated the pharmacological effects of cytochalasin and oryzalin on the directional movement and wiggling of mitochondria (Supple-mentary Fig. 4). Based on our observations, speed-based fre-quency analysis is useful for evaluating whether mitochondria

behave normally when treated with exogenous factors (e.g., che-micals and environmental changes).

Furthermore, the MSD analysis clearly separated wiggling as a short-distance migration with diffusive movement on chloroplast, while directional movement as a long-distance migration with high speed independently of chloroplast (Supplementary Figs. 11 and 12 and Supplementary Table 4). Both movements (wiggling and directional movement) are apparently different from Brow-nian motion. However, the MSD analysis of mitochondrial movement in cytochalasin-treated cell revealed low-diffusion coefficient and low velocity (Supplementary Figs. 11 and 12, and Supplementary Table 4), even when comparing with a mito-chondrial movement with a short-distance migration (<5 μm) and CA (Figs. 2, 4, and 7 and Supplementary Figs. 3, 4, and 10), suggesting that the wiggling may be influenced by F-actin.

It means that F-actin would contribute to extend migrate distance of the mitochondrial movement on chloroplast. Therefore, the wiggling does not represent thermal diffusion, but random motion including cytoskeleton related activity. While F-actin apparently has role in a long-distance migration of mitochondria in cytosol as component of actomyosin system.

In the present study, we determined that mitochondrial wiggling occurs in close proximity to or on chloroplasts (Figs. 7 and 8, Supplementary Figs. 8 and 9, and Supplementary Movies 10 and 11). In accordance with the number of PA and CA in mitochondrial movement (Fig. 7b), mitochondrial movements in PA and CA include large part of wiggling (low speed - high angle changes), whereas that in NA not (Fig. 7c). Moreover, disrupting F-actin with cytochalasin increased the number of wiggling mitochondria beside chloroplast (Figs. 4 and 5, Supplementary Figs. 3, 4, and 7, Supplementary Table 5, and Supplementary Movies 5–8). In this context, chloroplasts might inhibit the F-actin-dependent directional movement of mitochondria, inducing wiggling around chloroplasts for chloroplast – mitochondria interaction.

Moreover, western blot analysis of organellar proteins (mitochondria-localized MTS-Citrine and chloroplast-resident RuBisCO activase) showed that mitochondria were co-isolated with chloroplasts in the cytochalasin-treated leaves (Supplementary Fig. 15), indicating their interaction without F-actin. We further examined the interaction of mitochondrion with chloroplast by measuring a distance between them in mobile chloroplasts during time-lapse analysis. The result showed that the distances were kept stably under half width of chloroplast diameter even though chloroplast moved vigorously, meaning that the interaction tightly occurred (Supplementary Fig. 16 and Supplementary Movie 12). These biochemical and physiological analysis conformed that the interaction of wiggling mitochondria to chloroplasts is an F-actin independent event.

Overall, the wiggling is defined as a mitochondrial movement possessing a short-distance migration with lower speed and high angle changes associated with high diffusion and low mean velocity derived from the MSD analysis, which is induced by interacting with chloroplast independently of F-actin. The short-distance below 5 μm is related to chloroplast size, which mitochondria associate with. The directional movement is defined as a mitochondrial movement possessing a long-distance migration with high speed and low angle changes associated with low diffusion and high mean velocity derived from the MSD analysis, which depends on F-actin.

The mechanical force for interaction of wiggling mitochondria with chloroplasts is currently unclear. Given that wiggling of mitochondria is independent on F-actin, regulation of mitochondria wiggling is different from wiggling of Golgi body depend on fine F-actin[28] or short-actin polymerization on chloroplast surface[29]. Perhaps association between chloroplast and mitochondria would be mediated by a membrane contact site containing tethering factor(s). Although several tethering factors in plant organelles have been reported, such as the mitochondria associated-membrane contact between the endoplasmic reticulum and mitochondria[30], plastid associated membranes[31] between the endoplasmic reticulum and chloroplasts, and a mitochondrial transmembrane lipoprotein complex[31] between mitochondria and chloroplasts, tethering factors between chloroplasts and mitochondria remain to be identified. Identifying such tethering factors could provide clear evidence for the physiological importance of mitochondrial wiggling.

Physical interactions between mitochondria and chloroplasts are thought to be required to optimize metabolite exchange between pathways shared by these two organelles, including the alternative oxidase pathways, the malate/oxaloacetate shuttle, ascorbate transport, photorespiration, and lipid trafficking[4–6,30–33]. Given that the frequency of mitochondrial wiggling increases during association with chloroplasts, wiggling behavior might be a hallmark of metabolite exchange between mitochondria and chloroplasts. Furthermore, during physical interactions between mitochondria and chloroplasts, several metabolites, such as malate, NAD(P)H, ascorbate, $NH_3$, and ATP from the malate/oxaloacetate shuttle, transporters, or photorespiration mechanisms could be efficiently transported between mitochondria and chloroplasts[4–6]. Taken together, the two types of mitochondrial movement, wiggling and directional movement, would be related to energy supply and metabolic pathway among cellular compartments under the influence of cytoskeleton, cytoplasmic streaming, membrane transport, and undefined tethering factors in plant cells. The wiggling induced by association with chloroplast would have a meaning to affect chloroplast and cellular function.

Several methods have been developed for analyzing metabolite transport, including recent live-cell imaging technology to visualize metabolites using malic acid carbon dots[34] and various fluorescent protein-based NAD(P)H sensors[35] with super-resolution microscopy combined with various imaging methods such as fluorescence lifetime imaging microscopy[36], positron-emission tomography for $^{13}NH_3$[37], fluorescence resonance energy transfer-based methods for ATP[38], and Raman microscopy with chemical probes[39]. These imaging techniques might allow us to elucidate the mechanisms underlying the interactions and metabolite transport between mitochondria and chloroplasts.

## Methods

**Production of transgenic Arabidopsis thaliana**. To generate transgenic *Arabidopsis thaliana* (L.) Heynh. (Columbia) plants harboring mitochondria expressing Citrine yellow fluorescent protein, the binary vector pMpGWB106 harboring the AtSD3(1–50)-Citrine fusion gene MTS-Citrine[39] was transformed into *Agrobacterium tumefaciens* strain GV3101. After performing *Agrobacterium*-mediated transformation of *A. thaliana*, Citrine-positive, and hygromycin-resistant transformants ($T_3$) were selected. Transgenic *A. thaliana* (Columbia) plants containing fluorescent mitochondria expressing red fluorescent protein fused to the F1-ATPase delta-prime subunit (Mt-RFP) were kindly provided by Dr. Shin-ichi Arimura (The University of Tokyo, Japan)[22].

**Plant culture conditions**. Sterilized seeds were sown on a 0.8% agar plate containing 1% sucrose, 2.2 mg l$^{-1}$ Murashige and Skoog Basal Medium (Sigma-Aldrich, St. Louis, MO, USA), and 0.5 μg l$^{-1}$ MES-KOH (pH 5.7). Germination was induced at 4 °C in the dark for 48 h. The plants were grown in an incubator (Nihonika, Japan) at 22 °C under a 16 h white light/8 h dark cycle.

**Protoplast isolation and drug treatments**. Protoplasts were isolated from true leaves of 3-week-old plants using an enzyme cocktail containing 1.5% Cellulase R10 (Yakult Pharmaceutical Industry, Japan), 0.4% Macerozyme R10 (Yakult Pharmaceutical Industry), 0.4 M mannitol, 20 M KCl, 10 M CaCl$_2$, 20 M MES-KOH (pH 5.7), and 0.1% bovine serum albumin. The protoplasts were rinsed three times with buffer A containing 0.4 M mannitol, 70 M CaCl$_2$, and 5 mM MES-KOH (pH 5.7) and treated with various drugs diluted in 0.1% DMSO. DMSO (0.1%) was used as a control treatment to observe mitochondrial movement. To disrupt F-actin or microtubules, the protoplasts were incubated for 1 h in the light at 22 °C in 50 μM cytochalasin B or 10 μM oryzalin, respectively, prior to observation. The protoplasts were collected by centrifugation for 5 min at 50 × g, washed three times with buffer A, and used for CLSM analysis.

**Imaging analysis**. To track mitochondrial movement, CLSM analysis was performed using a Zeiss LSM880 equipped with a ×63 NA 1.4 oil immersion objective (Carl Zeiss, Germany). Citrine fluorescence and chlorophyll autofluorescence were observed with an excitation wavelength of 488 nm. The emission wavelength ranges for Citrine fluorescence and chlorophyll autofluorescence were 509–580 nm and 630–700 nm, respectively[40]. Time-lapse serial images of mitochondrial movement were obtained every 250 ms for 30 s with increasing size of a pinhole to 100 μm × 100 μm from 1 airy unit for tracking mitochondria in deep area. All mitochondrial movements in protoplasts were tracked using MTrackJ[41], a plugin of Fiji[42]. The brightness centroid of each mitochondrion was used to represent the mitochondrial position.

After tracking mitochondrial trajectories in the serial images, each value for mitochondrial position was acquired at coordinate ($x_n$ (μm), $y_n$ (μm)) $_{n=1–30}$, speed (μm s$^{-1}$), and angle change ($\Delta\theta$) at each time point. All sets of the speed and

the angle changes of each mitochondrion in 30 s were represented in a scatter plot of mitochondrial movement. The angle change is defined as the angular change between the most recent displacement vector[41]. The direct distance of mitochondrial movement was calculated between the first ($x_0$ (μm), $y_0$ (μm)) and last ($x_{30}$ (μm), $y_{30}$ (μm)) position. Tracking images were created by plotting all mitochondrial positions ($x_n$ (μm), $y_n$ (μm)) $_{n=1-30}$ within 30 s of mitochondrial movement.

To visualize F-actin, the pGWT35S-Lifeact-Citrine gene was transiently expressed in the Mt-RFP transgenic *A. thaliana* protoplasts via PEG-mediated gene transfer[43]. The organization of F-actin was observed in Mt-RFP protoplasts that had been treated with DMSO, 50 μM cytochalasin, or 500 μM cytochalasin for 2 h beginning at 6 h after transformation. The number of mitochondria with or without F-actin was counted in 10 protoplasts using the Cell Counter plugin in ImageJ[42]. Colocalization of F-actin and mitochondrion on chloroplast surface was analyzed based on the time-lapse images of the Mt-RFP protoplast expressing the Lifeact-Citrine. We characterized F-actin (green: Citrine fluorescence), mitochondrion (red: RFP fluorescence), and chloroplast (blue: chlorophyll fluorescence), and then determined their colocalization for over 12 s.

To determine whether mitochondria migrate via wiggling, we investigated the correlation between migration distance and interval length. The raw time-lapse footage contained 120 frames taken at 250 ms-intervals. To analyze the images at 5-s intervals, we chose frames 20, 40, 60, 80, 100, and 120 and viewed the discontinuous trajectories. The migration distances were calculated using MTrackJ.

To classify the dependence of mitochondrial movement on the association with chloroplasts, we analyzed the mitochondrial trajectories in the serial images. We classified all mitochondria into no, partial (more than 3 s), and continuous (within 30 s) association with chloroplasts.

To confirm the interaction between mitochondrion and chloroplast, a distance between centroid of both mitochondrion and chloroplast was measured during time-lapse analysis after treatment of 500 μM cytochalasin B. Each centroid was measured by a command Analyze in the Image J. We determined the interaction positively occurs if their distance was kept within half width of chloroplast diameter during the time-lapse analysis.

**Co-isolation of mitochondria and chloroplasts.** 4-week-old leaves from MTS-Citrine transgenic *A. thaliana* were submerged in 5 mL of 10 mM MES, pH 5.7, 10 mM MgCl₂ supplemented with DMSO, 50 μM, or 500 μM Cytochalasin B in 6-wells plates. The samples were incubated in an incubator (Nihonika, Japan) at 22 °C under 100 μmol m⁻² s⁻¹ light condition for 4 h. The interacting organelles were co-isolated using 40% Percoll gradient solution. The pre-treated samples were ground in 5 mL of organelle isolation solution containing 0.33 M sorbitol, 0.1 M Tris-HCl, pH 7.5, 0.1 M NaCl, 2 mM EDTA, and 1 mM MgCl₂ and 0.1% (w/v) BSA. The solution was passed through 2-layers of 22–25 nm Mirocloth (EMD Millipore Corporation, Billerica MA, USA) and overlaid on 40% (v/v) Percoll (Sigma-Aldrich, St. Louis MO, USA), in organelle isolation solution. The intact organelles were pelleted by centrifugation at $1700 \times g$ for 7 min at 4 °C. The organellar proteins were isolated from organelle pellet with 1x phosphate saline buffer (PBS), pH 7.4 + 0.1% (v/v) Triton-X100. Total leaf proteins were isolated with 1xPBS, pH 7.4, 0.1% Triton-X100, 2 mM EDTA, and 10% glycerol supplemented with Halt™ proteases inhibitor cocktail (Pierce Biotechnology, Rockford IL, USA). The organellar proteins (5 μg) or total leaf proteins (20 μg) were applied to western blotting using anti-GFP polyclonal antibody (NB600–308; Novus Bio-technology, Centennial CO, USA) or 1:5000 anti-rubisco activase (RA; chloroplast-protein marker) polyclonal antibody (AS10–700; Agrisera, Vännäs, Sweden). The horse reddish peroxidase-conjugated Goat anti-Rabbit IgG polyclonal antibody (Abcam, Tokyo, Japan) was used as a secondary antibody[44].

**Fixed-cell experiment.** To analyze fluctuations in fluorescence, 3-week-old leaves from MTS-Citrine transgenic *A. thaliana* were fixed with 4% paraformaldehyde and 2% glutaraldehyde in 0.1 M potassium phosphate buffer (pH 7.4) for 4 h at 4 °C. The samples were dehydrated in an ethanol series (25, 50, 75, and 90%), embedded in LR White resin (London Resin), and cut into semi-ultrathin sections (1 μm) using a PT-PC PowerTome Ultramicrotome (RMC Boeckeler) equipped with a DiATOME Histo diamond knife. The sections were collected onto coverslips and observed by CLSM. To observe Citrine fluorescence, we added 0.1 M sodium carbonate solution to the sections.

**MSD analysis.** Two-dimensional mean-squared displacement (MSD) analysis was performed following to the methods[21–25]. All mitochondrial positions ($x_n$ (μm), $y_n$ (μm)) $_{n=1-30}$ within 30 s of mitochondrial movement were obtained at every 1 s from three different protoplasts in each study based on the trajectory analysis using MTrackJ.

For finding the average mitochondrial squared displacement for every time interval of the trajectory, calculation of MSD were examined according to the equation (Eq. 1);

$$\text{MSD} = \Delta r^2(n\Delta t) = \frac{1}{N-n}\sum_{i=1}^{N-n}\left[(x_{i+n}-x_i)^2+(y_{i+n}-y_i)^2\right] \quad (1)$$

where $\Delta t$ is the time between frames (1 s in this study) and $n$ is the number of time

intervals ($n = 1, 2,\dots (N$ -1)). $N$ is a total number of images. Average MSD at each time points were plotted in Supplementary Figs. 11–13 and fitted to equations (Eqs. 2 and 3) as linear or curve model using the least-squares method (Supplementary Figs. 11–13).

$$\text{MSD}(\Delta t) = 4D\Delta t (\text{normal diffusion}) \quad (2)$$

where $D$ is a diffusion coefficient.

$$\text{MSD}(\Delta t) = v^2\Delta t^2 + 4D\Delta t (\text{directed motion with diffusion}) \quad (3)$$

where $v$ is a mean velocity and $D$ is a diffusion coefficient. The chi-squared value $\chi^2$ was used to test the goodness of fitting of the MSD analysis. The fitting to a null hypothesis indicates that the model fits the data. The high $p$-values does not mean rejecting the null hypotheses (Supplementary Table 4).

**Statistics and reproducibility.** For analyzing mitochondrial movements, the values were presented as mean ± SD from at least three biological replicates with more than 200 mitochondria and statistically compared using the unpaired Student's $t$ test. The exact number of biological replicates was provided in individual figure legends. The statistical analyses were performed using Excel (Microsoft(R) for Mac Ver. 16.16.18 (200112).

**Reporting summary.** Further information on research design is available in the Nature Research Reporting Summary linked to this article.

## Data availability

All relevant data are available from the authors upon request. Source data underlying plots shown in figures are provided in Supplementary Data I.

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

## Acknowledgements

We thank Ms. Naeko Shinozaki-Narikawa for kindly helping us with the fixed-cell experiment. We also thank Ms. Minami Shimizu, Dr. Kousuke Hanada, and Dr. Shinichi Arimura for kindly providing transgenic plants used to visualize mitochondria. This work was supported by the Japan Science and Technology Agency Exploratory Research for Advanced Technology program (JST-ERATO; grant number JPMJER1602).

## Author contributions

K.O., T.I., Y.K., and K.N. designed the study. K.O., T.I., C.T., C.H., K.T., Y.K., and K.N performed research. K.T., T.Y., and K.I. generated transgenic plants. K.T. and T.I. performed fixed-cell analysis. C.T. performed co-isolation assay. All the authors analyzed the data and wrote the manuscript.

## Competing interests

The authors declare no competing interests.
