## [Peer Review File · Communications Biology]

Reviewers' comments:

Reviewer #1 (Remarks to the Author):

This paper is based on carefully performed quantitative experiments on mitochondrial movement in plant protoplasts. While the authors have collected high quality data, their analysis is unconvincing as it stands and could be significantly improved.

1. The significance of the different types of mitochondrial motion is not clear. On the face of it, directional motion is associated with mitochondria being transported by motors of actin based mobility, while the other subset is associated with chloroplasts. Is this something that was not well understood in the plant biology field? The authors could improve the discussion and the introduction, specifically pointing out why the classification of these two types of mitochondria are of interest.

2. The classification of "wiggling" as opposed to "directional" is very ad hoc. There is a huge amount of work on this by now. While the authors are not physicists, there is no reason why they cannot analyze the mean squared displacement of the mitochondria and use that for the classification and the analysis. Please check out this paper for some ideas:

Particle tracking in living cells: a review of the mean square displacement method and beyond
Naama Gal · Diana Lechtman-Goldstein · Daphne Weihs
Rheol Acta (2013) 52:425–443

3. Previous work has shown that even seemingly random "wiggling" motion in cells has an active component. Analysis of the MSD may help in figuring out whether this active component is present in the mitochondria that associate with chloroplasts.

4. The results of the paper suggest that mitochondria that are in association with chloroplasts are not connected to active molecular motors and may be tethered to the chloroplasts. Comparison of the MSD between cytochalasin treated cells and WT cells will help in figuring out whether the actin cytoskeleton or microtubules serves to confine the "wiggling" population or not. The videos seem to suggest that this is what is happening.

I recommend re-analysis of all the results based on the MSD of the mitochondria.

Optional: Even more insight is obtained by analyzing the distribution of directional persistence. Please see: Burov et. al. : Distribution of directional change as a signature of complex dynamics, PNAS vol 110, pp. 19689 (2013)

Reviewer #2 (Remarks to the Author):

Oikawa et al. described dynamic movements of plant mitochondria in mesophyll protoplasts and leaf mesophyll cells of *A. thaliana* with live-cell imaging technology. The puzzled moving paths of mitochondria were divided into two distinct types, long distance directional movement and short distance wiggling movement. On the basis of pharmacological analysis, they suggested that the directional movement is mediated in a F-actin-dependent manner while the wiggling movement is not related with F-actin. The manuscript addresses an interesting topic, however the manuscript is not supported fully by the presented data.

Specific points regarding the experiments:

The authors interpret that wiggling movement is mediated in a f-actin independent manner. However, the movements are also found on the continuous line of directional movement around/in the chlorophyll autofluorescence, and vice versa (Supplementary movie 1). It does not rule out the possibility that wiggling movement could be mediated by chloroplasts (as physical barriers) on the moving paths of mitochondria along f-actin.

Fig 2c and Fig 3. The authors represent the data in two groups: low speed with high angle change (gray region) and high speed with low angle change (pale-blue region). It is hard to follow the interpretation. First, what does angle change mean? Please describe detailed method to measure angles.

The low speed population exhibit a broad range of angles indicating that the mitochondria movements are not homogeneously regulated as Brownian movement. Actually, a few populations of mitochondria (NA in Fig. 7) exhibit high speed and low angle change and most mitochondria exhibit low speed and random movement.

Chloroplasts are associated with the plasma membrane (Oikawa et al., 2003 Plant Cell; Oikawa et al., 2008 Plant Physiology). Mitochondria with directional movement could be hindered by chloroplasts. Therefore, to insist the F-actin-independent interaction of mitochondria with chloroplasts should be confirmed by other experiments such as biochemical analysis and chloroplast movement.

Fig 4c. The chloroplast is abnormal in the size ($>20 \mu\text{m}$) in comparison with those presented in other Figs ($\sim 10 \mu\text{m}$). It is hard to recognize the MTrackJ line. Please use the thicker line.

Fig 5. The data is not sufficient to confirm the associations of mitochondria with chloroplast and F-actin. In addition, how are the wiggling mitochondria defined in Fig. 5? The quantitative data should be included in Fig 5.

Line 194-196, It is hard to understand the meaning. Please rewrite it.

Line 200-202, The data is not meaningful because the range of data is less than the resolution of confocal microscopy.

Minor points:

1. Please check the positions of $>5 \mu\text{m}$ and $<5 \mu\text{m}$ in Fig 3b
2. In Supplementary Fig. 6, Constant association (CA) \diamond Continuous association.

Reviewer #3 (Remarks to the Author):

This paper describes the different types of motility that are exhibited by mitochondria, changes in this behavior when in the vicinity of a chloroplast and the effect of actin and microtubule depolymerisation on motility of mitochondria.

When I read the manuscript I was surprised that many relevant papers that (in part) come to similar conclusions as in this paper were not cited. For example, the work of the Mathur group is completely missing (see e.g. Barton et al., JCS 2018), no literature is cited that covers the specific organisation and function of the peri-plastidal actin cytoskeleton (for review see e.g. Wada, Plant Sci. 2013) and Akkerman et al., Plant Cell Physiol 2011. There are significant overlaps between the work and conclusions in the current manuscript and the literature that is not cited.

Besides this problem, there are multiple issues, some of which are mentioned below:

Why were protoplasts taken as a model system? The authors repeat the experiments in intact leaf mesophyll cells, but the main results are obtained in protoplasts. It is uncertain how the absence of a cell wall or the removal from a tissue context affect the intracellular organisation or motility.

I presume that single z-plane time series were collected. How do the authors consider mitochondria that move in or out of the focal plane during their analyses?

The conclusion that microtubules do not affect the movement of mitochondria is premature. Since microtubules are likely to have a cortical localisation, only the motility in this plane should be considered. To link motility to microtubules, co-localisation of mitochondria and microtubules would be essential.

How do the authors define association with a chloroplast? An attachment is not shown, but it depends on the definition how this is interpreted. Since it is likely that these protoplasts have large central vacuoles, there may be a thin layer of cytoplasm that surrounds plastids. Mitochondria trapped in this cytoplasm may not necessarily be associated with a chloroplast. Besides this problem, the mitochondria in the vicinity of a chloroplast show a similar wiggling behavior as the mitochondria that are not associated (see e.g. movie 4). In the absence of actin, something needs to generate a force for directional movement. That might be a flux in the cytoplasm, perhaps originating from membrane transport from plastid to cytoplasm and vice versa.

I would love to see decent statistics on the performed analyses.

Response to Reviewer #1

We sincerely express our appreciation to the Reviewer #1 for giving us valuable comments and suggestions, which have improved our manuscript. We respond to the reviewer's comments point by point as follows:

Reviewer #1 (Remarks to the Author)

This paper is based on carefully performed quantitative experiments on mitochondrial movement in plant protoplasts. While the authors have collected high quality data, their analysis is unconvincing as it stands and could be significantly improved.

Comment 1. The significance of the different types of mitochondrial motion is not clear. On the face of it, directional motion is associated with mitochondria being transported by motors of actin based mobility, while the other subset is associated with chloroplasts. Is this something that was not well understood in the plant biology field? The authors could improve the discussion and the introduction, specifically pointing out why the classification of these two types of mitochondria are of interest.

Response 1

We thank the reviewer for providing the fundamental and significant comments on our study. We improved the introduction and the discussion to clear the importance of classification of directional movement and wiggling, based on the reviewer's comments. The interesting point of the classification of two mitochondrial movements is that directional movement depends on cytoskeleton, whereas wiggling depends on interaction with chloroplast. These different types of mitochondrial movements are related to different cellular mechanisms and functions such as metabolic pathway and energy supply. These are significant points, however, it has not well been characterized in plant biology field. As the reviewer pointed out, F-actin-dependent-mitochondrial movement has been well characterized in plant cells (Logan and Leaver, 2000⁸; Van Gestel et al, 2002¹¹; Sheahan et al, 2005¹⁴; Doniwa et al, 2007¹⁵; Zheng et al, 2009¹², 2010¹³) and the wiggling in F-actin-disrupted plant cells has also been reported (Sheahan et al, 2005¹⁴; Zheng et al, 2009¹²). However, the meaning and mechanism of the wiggling, which is induced by associating with chloroplast in normal leaf mesophyll cell, has not been clarified in detail to date. To explain these backgrounds, we inserted the additional sentences in Introduction and Discussion sections as follows: (Page 4, Lines 61-63)

“This report suggested that mitochondrial movements would be influenced by chloroplast¹⁶. However, to date, how the mitochondrial movement is influenced by its association with chloroplast has not been clarified.”

(Page 31, Lines 437-441)

“Taken together, the two types of mitochondrial movement, wiggling and directional movement,

would be related to energy supply and metabolic pathway among cellular compartments under the influence of cytoskeleton, cytoplasmic streaming, membrane transport, and undefined tethering factors in plant cells. The wiggling induced by association with chloroplast would have an significant meaning to affect chloroplast and cellular function.”

Comment 2. The classification of “wiggling” as opposed to “directional” is very ad hoc. There is a huge amount of work on this by now. While the authors are not physicists, there is no reason why they cannot analyze the mean squared displacement of the mitochondria and use that for the classification and the analysis. Please check out this paper for some ideas: Particle tracking in living cells: a review of the mean square displacement method and beyond Naama Gal · Diana Lechtman-Goldstein · Daphne Weihs Rheol Acta (2013) 52:425–443.

Response 2

We thank the reviewer for suggesting that the mean squared displacement (MSD) analysis should be applied to this study. We examined MSD analysis of the mitochondrial movements for the classification following to the reviewer’s comments (Pages 38-39, Lines 549-569; Methods for MSD analysis). We calculated the two-dimensional MSD $\langle r^2 \rangle$ for each trajectory of the mitochondrial movements (Gal et al. 2013²¹; Saxton & Jacobson, 1997²³). At first, we fitted the MSD plot to an equation (Eq 1):

$$\langle r^2 \rangle = 4Dt^\alpha \quad (\text{Eq 1})$$

where D is a diffusion coefficient, t is the time between frames, and α is the MSD scaling exponent ($0 \leq \alpha < 2$). However α value of the MSD in the oryzalin-treated cell and PA (partially associating with chloroplast) exhibited over ballistic limitation ($2 < \alpha$) (Gal et al. 2013²¹). The results indicated that the Eq 1 wasn’t fitted to the MSD plots. Therefore, we fitted the MSD plots to an Eq 2:

$$\langle r^2 \rangle = v^2 \Delta t^2 + 4D\Delta t \quad (\text{Eq 2})$$

where v is a mean velocity, Δt is the time between frames, and D is a diffusion coefficient (Supplementary Figs. 10,11, and 12; Supplementary Table 4 attached below) (Kusumi et al, 1993²²; Saxton & Jacobson, 1997²³). The results of the MSD analysis are explained in the next response (Response 3).

Supplementary Fig. 10 MSD analysis of mitochondrial movement. **a-f** Each plot is fitted to linear or curve models using the least-squares method with the Eq 2. **a, b** The mean squared displacement (MSD) of mitochondrial movements, which are separated to shorter (open circles) or longer (filled-magenta circles) than 5.0 μm distance. **c-e** MSD analysis for mitochondrial movement in DMSO-, oryzalin-, and cytochalasin-treated cells. **f** MSD analysis for mitochondria movement in fixed cell.

Supplementary Fig. 11 MSD analysis of mitochondrial movement associating with or without chloroplast. Each plot is fitted to curve models using the least-squares method with the Eq 2. **a-c** The mean squared displacement (MSD) analysis of three types of mitochondrial movements; no association (NA; **a**), partial association (PA; **b**), and continuous association (CA; **c**) with chloroplasts.

Supplementary Fig. 12 MSD analysis of mitochondrial movement in oryzalin-treated cell. Each plot is fitted to curve models using the least-squares method with the Eq 2. The mean squared displacement (MSD) of mitochondrial movement in oryzalin-treated cells, which are separated to shorter (a) or longer (b) than 5.0 μm -migrate distance.

Supplementary Table 4

Parameter of MSD analysis and character of mitochondrial movements

	D ($\mu\text{m}^2/\text{s}$)	v ($\mu\text{m}/\text{s}$)	Pattern
MD < 5 μm	0.11	0.0332	Dire + Diff
5 μm < MD	-	0.406	Dire
DMSO	-	0.263	Dire
Oryzalin	-	0.289	Dire
• MD < 5 μm (Oryzalin)	0.038	0.134	Dire + Diff
• 5 μm < MD (Oryzalin)	-	0.475	Dire
Cytochalasin	0.00010	0.0100	Dire + Diff
NA	0.00075	0.254	Dire + Diff
PA	-	0.267	Dire
CA	0.029	0.0436	Dire + Diff
Fixed	0.00023	-	Brownian

D : Diffusion coefficient, v : mean velocity, MD: migrate distance, NA: No association with chloroplast, PA: partial association with chloroplast, CA: continuous association with chloroplast, Dire: Direct, and Diff: Diffusion.

Comment 3. Previous work has shown that even seemingly random “wiggling” motion in cells has an active component. Analysis of the MSD may help in figuring out whether this active component is present in the mitochondria that associate with chloroplasts.

Response 3

We could separate the wiggling, directional movement, and Brownian motion using MSD analysis. The MSD of mitochondrial movements with both short-distance migration ($< 5 \mu\text{m}$) and the continuous associating with chloroplast group (CA) revealed directed- and diffusive motion with high-coefficient value (D), whereas mitochondrial movements with long-distance migration ($5 \mu\text{m} <$), in oryzalin-treated cell, the no associating (NA) and partially associating (PA) with chloroplast group showed directed motion with high-value of the mean v with or without low value of the D . The $5 \mu\text{m}$ was determined as the threshold from which the second peak appeared in distribution of the migrate distances in Fig. 2b and Supplementary Fig. 1. We thought that $5 \mu\text{m}$ is related to chloroplast diameter (described in Discussion; Page 25, Lines 328-331). We added Supplementary Figures 10 and 11, and Supplementary Table 4 as shown above (Response 2). We concluded that the directed and diffusive motion with high value of the diffusion coefficient is a future of the wiggling defined as active components. To explain the results of the MSD analysis which defined the wiggling as a mitochondrial movement associating with chloroplast, we inserted additional sentences in Results and Discussion as follows:

(Pages 24-25, Lines 303-320)

“To confirm characteristics of mitochondrial movements, we performed mean-squared displacement (MSD) analysis^{21,22,23,24} (Eq 1; see MSD analysis in Methods) on the trajectories of each mitochondrial movement in Figs. 2c, 4c, 6d, e, and 7c. The two-dimensional MSD plots revealed liner or parabolic shape, and were fitted to Eq 2 or Eq 3 (see MSD analysis in Methods, Supplementary Figs. 10-12, Supplementary Table 4). The fixed cell revealed liner shape defined as Brownian motion with low diffusion (D ; 0.00023)^{23,24,25} (Supplementary Fig. 10f). Mitochondrial movement with long-distance migration ($5 \mu\text{m} <$), DMSO-treatment, oryzalin-treatment, NA, and PA exhibiting directed diffusion curve with steeper sloop, meaning directed motion with high velocity (v : 0.406, 0.289, 0.263, 0.254, and 0.267, respectively) (Supplementary Figs. 10b, c, d and 11a, b). On the other hand, mitochondrial movement with short-distance migration ($< 5 \mu\text{m}$) and CA revealed both directed- and diffusive motion with slow slope (D : 0.11 and 0.029, respectively) (v : 0.0332 and 0.0436, respectively) (Supplementary Figs. 10a and 11c, Supplementary Table 4). Mitochondrial movement with cytochalasin-treatment also revealed both directed- and diffusive motion with slow slope, but low velocity (v : 0.0100) and low diffusion (D : 0.00010) (Supplementary Fig. 10e, Supplementary Table 4). These MSD results confirmed that the wiggling has diffusive motion with low velocity, and that directional movement has high velocity.”

(Page 28, Lines 375-379)

“Furthermore, the MSD analysis clearly separated wiggling as a short-distance migration with diffusive movement on chloroplast, while directional movement as a long-distance migration with high speed independently of chloroplast (Supplementary Figs. 10 and 11, Supplementary Table 4). Both movements (wiggling and directional movement) are apparently different from Brownian motion.”

Taken together with the analysis of the trajectory, the MSD, biochemical co-precipitation, and physiological interaction, we redefined the wiggling and directional movement as follows:

(Page 30, Lines 408-415)

“Overall, the wiggling is defined as a mitochondrial movement possessing a short-distance migration with lower speed and high angle changes associated with high diffusion and low mean velocity derived from the MSD analysis, which is induced by interacting with chloroplast independently of F-actin. The short-distance below 5 μm is related to chloroplast size, which mitochondria associate with. The directional movement is defined as a mitochondrial movement possessing a long-distance migration with high speed and low angle changes associated with low diffusion and high mean velocity derived from the MSD analysis, which depends on F-actin.”

Comment 4. The results of the paper suggest that mitochondria that are in association with chloroplasts are not connected to active molecular motors and may be tethered to the chloroplasts. Comparison of the MSD between cytochalasin treated cells and WT cells will help in figuring out whether the actin cytoskeleton or microtubules serves to confine the “wiggling” population or not. The videos seem to suggest that this is what is happening. I recommend re-analysis of all the results based on the MSD of the mitochondria. Optional: Even more insight is obtained by analyzing the distribution of directional persistence. Please see: Burov et. al. : Distribution of directional change as a signature of complex dynamics, PNAS vol 110, pp. 19689 (2013)

Response 4

Following to the reviewer’s suggestion, we performed the MSD analysis of mitochondrial movements in cytochalasin- and oryzalin-treated protoplasts (i.e., F-actin- and microtubules-disrupted cells, respectively), and compared the MSD patterns with the MSD pattern of mitochondrial movements in CA. By the MSD analysis, we could reveal that CA was independent of actin cytoskeleton or microtubule.

In cytochalasin-treated cells (F-actin-disrupted cells), the MSD pattern of the mitochondrial movements revealed both directed- and diffusive motion (Supplementary Fig. 10e), similar to that in CA (Supplementary Fig. 11c; Supplementary Table 4). However, both D and v values were lower in cytochalasin-treated cell than those in CA. Therefore, we concluded that the wiggling was induced independently of F-actin, while migration distance of the mitochondrial movement on chloroplast would

be expanded by F-actin. We added the explanations for the contribution of F-actin to mitochondrial movements in Result as follows:

(Pages 28-29, Lines 379-388)

“However, the MSD analysis of mitochondrial movement in cytochalasin-treated cell revealed low-diffusion coefficient and low velocity, meaning that mitochondria exhibited diffusion in short range in F-actin free condition (Supplementary Fig. 10e). It indicated that wiggling is independent of F-actin, but F-actin would contribute to extend migrate distance of the mitochondrial movement on chloroplast, because speed of mitochondrial movement in cytochalasin-treated cell was dramatically reduced, as compared to a mitochondrial movement with a short-distance migration (less than 5 μm) and CA (Figs. 2, 4, and 7, Supplementary Figs. 2 and 3). While F-actin apparently has role in a long-distance migration of mitochondria in cytosol as component of actomyosin system.”

In the case of the oryzalin-treated cells (microtubules-disrupted cells), the MSD of the mitochondrial movement with a short-distance migration ($<5 \mu\text{m}$) revealed a similar pattern to that of the wiggling, which has high value of D and low value of v (Supplementary Fig. 12; Supplementary Table 4 in Response 2). Therefore, we concluded that microtubule doesn't serve to induce the wiggling on chloroplast. We inserted sentences about the contribution of microtubule to the wiggling as follows:

(Page 27, Lines 360-364)

“In the current study, the MSD analysis showed that mitochondrial movement with a short-distance migration ($< 5 \mu\text{m}$) in oryzalin-treated cell maintained both directed- and diffusive motion with slow slope (D : 0.038, v : 0.134) (Supplementary Fig. 12, Supplementary Table 4), which is characteristic of the mitochondrial wiggling. Thus, it appears that microtubules are not a driving force for the mitochondrial wiggling.”

Response to Reviewer #2

We sincerely thank to Reviewer #2 for giving us insightful comments and suggestions, which have improved our manuscript. We have carefully addressed to the reviewer's comments and revised the manuscript as follow:

Reviewer #2 (Remarks to the Author)

Oikawa et al. described dynamic movements of plant mitochondria in mesophyll protoplasts and leaf mesophyll cells of *A. thaliana* with live-cell imaging technology. The puzzled moving paths of mitochondria were divided into two distinct types, long distance directional movement and short distance wiggling movement. On the basis of pharmacological analysis, they suggested that the directional movement is mediated in a F-actin-dependent manner while the wiggling movement is not related with F-actin. The manuscript addresses an interesting topic, however the manuscript is not supported fully by the presented data.

Specific points regarding the experiments:

Comment 1. The authors interpret that “Wiggling” movement is mediated in a f-actin independent manner. However, the movements are also found on the continuous line of directional movement around/in the chlorophyll autofluorescence, and vice versa (Supplementary movie 1). It does not rule out the possibility that “Wiggling” movement could be mediated by chloroplasts (as physical barriers) on the moving paths of mitochondria along f-actin.

Response 1

We thank the reviewer for providing the insight about the wiggling movement is mediated by chloroplasts as simple physical barriers. Supplementary Movie 1 shows the details of the mitochondrial movements, which the reviewer pointed out. The mitochondria associating with chloroplasts revealed both the wiggling or liner movement on chloroplast before left it. However, we thought that the liner mitochondrial movements on chloroplast were not the wiggling defined in the present study. We would like to explain the reason to rule out the possibility that the reviewer suggested.

We defined the wiggling as a mitochondrial movement that consists of low speed with high angle changes, such as a short-distance migration (less than 5 μm in 30 s) on chloroplast. The 5 μm was determined as the threshold of the migration length (Fig. 2b; Supplementary Fig. 1), related to chloroplast diameter and association with chloroplast as described in Discussion (Page 25, Lines 328-331). We inserted the sentence about the definition of the wiggling in Results as follows:

(Pages, 9-10, Lines 133-138)

“Taken together, mitochondria migrating less than 5 μm in 30 s moved at low speed with high angle change rates defined as wiggling, and mitochondria migrating more than 5 μm in 30 s moved at high speed with low angle change rates defined as directional movement. These results indicate that our method for evaluating and quantifying mitochondrial movements was sufficient to further explore

the differences between wiggling and directional movement.”

We showed that the wiggling still occurred and increased in F-actin disrupted-cells (Figs. 4, 5, and 6, Supplementary Movies 4 and 5). In the revised manuscript, by the MSD analysis, we confirmed to clearly separate the wiggling from the F-actin-dependent directional movement and concluded that the wiggling is independent of F-actin. Based on the results, we rule out the possibility that wiggling movement could be mediated by F-actin along chloroplasts. We inserted the sentences about the MSD analysis in Results and Figures 10, 11, and Supplementary Table 4 as follows:

(Pages 28-29, Lines 375-388)

“Furthermore, the MSD analysis clearly separated wiggling as a short-distance migration with diffusive movement on chloroplast, while directional movement as a long-distance migration with high speed independently of chloroplast (Supplementary Figs. 10 and 11, Supplementary Table 4). Both movements (wiggling and directional movement) are apparently different from Brownian motion. However, the MSD analysis of mitochondrial movement in cytochalasin-treated cell revealed low-diffusion coefficient and low velocity, meaning that mitochondria exhibited diffusion in short range in F-actin free condition (Supplementary Fig. 10e). It indicated that wiggling is independent of F-actin, but F-actin would contribute to extend migrate distance of the mitochondrial movement on chloroplast, because speed of mitochondrial movement in cytochalasin-treated cell was dramatically reduced, as compared to a mitochondrial movement with a short-distance migration (less than 5 μm) and CA (Figs. 2, 4, and 7, Supplementary Figs. 2 and 3). While F-actin apparently has role in a long-distance migration of mitochondria in cytosol as component of actomyosin system.”

Supplementary Fig. 10 MSD analysis of mitochondrial movement. **a-f** Each plot is fitted to linear or curve models using the least-squares method with the Eq 2. **a, b** The mean squared displacement (MSD) of mitochondrial movements, which are separated to shorter (open circles) or longer (filled-magenta circles) than 5.0 μm distance. **c-e** MSD analysis for mitochondrial movement in DMSO-, oryzalin-, and cytochalasin-treated cells. **f** MSD analysis for mitochondria movement in fixed cell.

Supplementary Fig. 11 MSD analysis of mitochondrial movement associating with or without chloroplast. Each plot is fitted to curve models using the least-squares method with the Eq 2. **a-c** The mean squared displacement (MSD) analysis of three types of mitochondrial movements; no association (NA; **a**), partial association (PA; **b**), and continuous association (CA; **c**) with chloroplasts.

Supplementary Table 4 Parameter of MSD analysis and character of mitochondrial movements

	D ($\mu\text{m}^2/\text{s}$)	v ($\mu\text{m}/\text{s}$)	Pattern
MD < 5 μm	0.11	0.0332	Dire + Diff
5 μm < MD	-	0.406	Dire
DMSO	-	0.263	Dire
Oryzalin	-	0.289	Dire
• MD < 5 μm (Oryzalin)	0.038	0.134	Dire + Diff
• 5 μm < MD (Oryzalin)	-	0.475	Dire
Cytochalasin	0.00010	0.0100	Dire + Diff
NA	0.00075	0.254	Dire + Diff
PA	-	0.267	Dire
CA	0.029	0.0436	Dire + Diff
Fixed	0.00023	-	Brownian

D : Diffusion coefficient, v : mean velocity, MD: migrate distance, NA: No association with chloroplast, PA: partial association with chloroplast, CA: continuous association with chloroplast, Dire: Direct, and Diff: Diffusion.

Comment 2. Fig 2c and Fig 3. The authors represent the data in two groups: low speed with high angle change (gray region) and high speed with low angle change (pale-blue region). It is hard to follow the interpretation. First, what does angle change mean? Please describe detailed method to measure angles.

Response 2

We thank the reviewer for pointing out our obscurity in Fig. 2c and Fig. 3. We defined “angle change” as the angular change between the most recent displacement vectors in trajectory analysis. The speed corresponds to the migrate distance between the two points of the vectors taken at every 1 s. These data were obtained from the trajectory analysis of the time-lapse images by MTrackJ, as described in the experimental section. To clear the confusions, we inserted the additional sentences about the angle change and the speed to Methods as follows:

(Pages 34-35, Lines 483-492)

“Time-lapse serial images of mitochondrial movement were obtained every 250 ms for 30 s with increasing size of a pinhole to 100 μm x 100 μm from 1 airy unit for tracking mitochondria in deep area. All mitochondrial movements in protoplasts were tracked using MTrackJ⁴¹, a plugin of Fiji⁴². The brightness centroid of each mitochondrion was used to represent the mitochondrial position.

After tracking mitochondrial trajectories in the serial images, each value for mitochondrial position was acquired at coordinate $(x_n (\mu\text{m}), y_n (\mu\text{m}))_{n=1-30}$, speed $(\mu\text{m s}^{-1})$, and angle change $(\Delta\theta)$ at each time point. All sets of the speed and the angle changes of each mitochondrion in 30 s were represented in a scatter plot of mitochondrial movement. The angle change is defined as the angular change between the most recent displacement vector⁴¹.”

To clarify “two groups: low speed with high angle change (gray region) and high speed with low angle change (pale-blue region)” as the reviewer pointed out, we classified mitochondrial movements based on criteria as follows. At first, we separated mitochondria by the migrate distance obtained from trajectory analysis of mitochondrial movements at every 1 s for 30 s (Fig. 2b; Supplementary Fig. 1). Two different colors of plots in Fig. 2c mean different categories based on mitochondrial migration distance of less or more than 5 μm in 30 s. The 5 μm was determined as the threshold from which the second peak appeared in the distribution of the migrate distances (Fig. 2b; Supplementary Fig. 1). As described above, we thought that the 5 μm is related to chloroplast diameter. Secondly, we plotted all sets of the speed and the angle changes, which each mitochondrial movement possessed. We separated two different types of mitochondrial movements according to distribution patterns of scatter plot (low speed with high-angle changes; gray region, and high speed with low angle changes; pale-blue region) (Fig. 2c). The result corresponded to the migrate distance. We showed each representative image (Fig. 3) derived from Fig. 2c. Lastly, we further performed the MSD analysis and could separate two regions clearly. The wiggling has directed- and high diffusive components, whereas directional movement has only a directed component (high mean velocity). To explain these criteria to distinguish those two groups, we inserted the sentences about the MSD analysis to clear the wiggling from other mitochondrial movements as follows:

(Page 28, Lines 375-379)

“Furthermore, the MSD analysis clearly separated wiggling as a short-distance migration with diffusive movement on chloroplast, while directional movement as a long-distance migration with high speed independently of chloroplast (Supplementary Figs. 10 and 11, Supplementary Table 4). Both movements (wiggling and directional movement) are apparently different from Brownian motion.”

Comment 3. The low speed population exhibit a broad range of angles indicating that the mitochondrial movements are not homogeneously regulated as Brownian movement. Actually, a few populations of mitochondria (NA in Fig. 7) exhibit high speed and low angle change and most mitochondria exhibit low speed and random movement.

Response 3

In accordance with the reviewer’s comment, mitochondrial movements of low speed ($< 0.4 \mu\text{m/s}$) have a broad range of angle change (Fig. 7c). The MSD analysis revealed that mitochondrial movement were

independent of Brownian movement, as described above. The scatter plot for the speed and angle changes of mitochondrial movements in mitochondria continuously associating with chloroplast (CA) exhibited mostly with low speed - high angle changes (Fig.7c, CA). These results depended on how long mitochondria associated with chloroplasts. As described above, we examined the MSD analysis of mitochondria movements in NA, partially associating with chloroplast group (PA), and CA, and clearly separated that in CA from the others (Supplementary Fig.11 in Response 1). These movements were different from Brownian movement (Supplementary Fig.10f). The MSD of mitochondrial movements in NA and PA revealed directed motion with high value of the v , differing from that in CA, which has both directed- and diffusive motion with high value of the D . We inserted the sentence about the MSD analysis of mitochondria movement in NA, PA, and CA in Result as follows:

(Pages 24-25, Lines 309-320)

“Mitochondrial movement with long-distance migration ($> 5 \mu\text{m}$), DMSO-treatment, oryzalin-treatment, NA, and PA exhibiting directed diffusion curve with steeper slope, meaning directed motion with high velocity (v : 0.406, 0.289, 0.263, 0.254, and 0.267, respectively) (Supplementary Figs. 10b, c, d and 11a, b). On the other hand, mitochondrial movement with short-distance migration ($< 5 \mu\text{m}$) and CA revealed both directed- and diffusive motion with slow slope (D : 0.11 and 0.029, respectively) (v : 0.0332 and 0.0436, respectively) (Supplementary Figs. 10a and 11c, Supplementary Table 4). Mitochondrial movement with cytochalasin-treatment also revealed both directed- and diffusive motion with slow slope, but low velocity (v : 0.0100) and low diffusion (D : 0.00010) (Supplementary Fig. 10e, Supplementary Table 4). These MSD results confirmed that the wiggling has diffusive motion with low velocity, and that directional movement has high velocity.”

Comment 4. Chloroplasts are associated with the plasma membrane (Oikawa et al., 2003 Plant Cell; Oikawa et al., 2008 Plant Physiology). Mitochondria with directional movement could be hindered by chloroplasts. Therefore, to insist the F-actin-independent interaction of mitochondria with chloroplasts should be confirmed by other experiments such as biochemical analysis and chloroplast movement.

Response 4

We appreciate the reviewer's comment for additional experiments to confirm the F-actin-independent interaction between mitochondria and chloroplasts. To elucidate the interaction, we examined both biochemical and physiological approaches. At first, we examined biochemically co-isolation assay, revealing that mitochondria were detected in chloroplast fraction isolated by centrifugation using Percoll gradient (Co-isolation of mitochondria and chloroplasts in Methods; Pages 37-38, Lines 520-538). We inserted the sentences about co-isolation assay and Supplementary Figure.14 as follows:

(Page 29, Lines 398-401)

“Moreover, western blot analysis of organellar proteins (mitochondria-localized MTS-Citrine and

chloroplast-resident RuBisCO activase) showed that mitochondria were co-isolated with chloroplasts in the cytochalasin-treated leaves (Supplementary Fig. 14), indicating their interaction without F-actin.”

Supplementary Fig. 14 Co-isolation analysis of mitochondria and chloroplasts. a, b Western blot analysis of the isolated-chloroplast fraction (left) or total-leaf extract (right) using antibodies against GFP (a) or RuBisCO activase (RA) (b). MTS-Citrine at 28kD (a) and RA at 45kD (b) (arrow heads) are detected in both CP fraction and total leaf extraction from 0 μM , 50 μM and 500 μM cytochalasin-treated cells. c, d The images (c, d) show the equal loading of western blotting proteins (a, b) after transferred onto the PVDF membrane and stained with the Ponceau S solution.

Next, we examined the direct interaction between mitochondrion and chloroplast focusing on chloroplast movement in F-actin-disrupted cell. Without F-actin, usual light- and actin-dependent chloroplast movement stopped, however chloroplasts moved vigorously with mitochondria in accordance with protoplast. We measured the distance between a centroid of chloroplast and mitochondrion during time-lapse analysis (Methods; Page 36, Lines 514-518), revealing that interaction between two organelles

was stable. We showed physiologically that interaction between mitochondrion and chloroplast was F-actin-independent. We inserted the sentences about the physiological experiment in Results as follows: (Pages 29-30, Lines 401-405)

“We further examined the interaction of mitochondrion with chloroplast by measuring a distance between them in mobile chloroplasts during time-lapse analysis. The result showed that the distances were kept stably under half width of chloroplast diameter even though chloroplast moved vigorously, meaning that the interaction tightly occurred (Supplementary Fig. 15, Supplementary Movie 9).”

a

b

Supplementary Fig. 15 Interaction between mitochondrion and chloroplast in F-actin-disrupted protoplasts.

a Representative image of trajectories of mitochondria (magenta) and chloroplast (blue) in a cytochalasin-treated protoplast, which expresses pGWT35S-Lifeact-Citrine gene (green). Time-lapse images were acquired for 30 s at 1s intervals. Trajectory within 30 s of centroid of chloroplast (C) and mitochondrion (M) are shown as red color. Scale bar; 5 μm . **b** Distance between centroid of mitochondria and chloroplasts at each time point in 30 s are shown as lines. Each three set of chloroplast and mitochondrion (1-3) from three protoplasts (No. 1-3) is shown as different colors and lines (No. 1-1 to No. 3-3).

Comment 5. Fig 4c. The chloroplast is abnormal in the size ($>20\ \mu\text{m}$) in comparison with those presented in other Figs ($\sim 10\ \mu\text{m}$).

Response 5

We thank the reviewer for pointing out the abnormal size of chloroplasts in Fig 4b (Fig 4c in the previous manuscript). We corrected the size of scale bar to $2\ \mu\text{m}$ in the Fig. 4b.

Fig. 4 Characterization of mitochondrial movements in the presence of cytoskeletal inhibitors. **a** Trajectories of mitochondrial movements in DMSO-, cytochalasin B (cytochalasin)-, and oryzalin-treated protoplasts acquired from time-lapse analysis of images taken for 30 s at 250-ms intervals. Scale bar; $5\ \mu\text{m}$. **b** Time-lapse images of mitochondria on the chloroplast surface in cytochalasin-treated protoplasts at 3-s intervals. Scale bar: $2\ \mu\text{m}$. **c** Scatter plot of speed (x-axis) and angle changes (y-axis) of mitochondria at each time point acquired from the trajectories of mitochondrial movements in (a). Protoplasts treated with DMSO (blue open circles), cytochalasin (red open circles), and oryzalin (green open circles).

Comment 6. It is hard to recognize the MTrackJ line. Please use the thicker line.

Response 6

We modified the lines to be clearer in Fig. 4b (Fig. 4c in the previous manuscript) put in the Response 5.

Comment 7. Fig 5. The data is not sufficient to confirm the associations of mitochondria with chloroplast and F-actin.

Response 7

We thank the reviewer for providing us insufficient and unclear points. We confirmed the interaction between mitochondrion and F-actin on chloroplast, if the interaction was kept for more than 12 s during the time-lapse analysis. We inserted the sentence about how we determined the interaction in Methods as follows:

(Pages 35-36, Lines 501-505)

“Colocalization of F-actin and mitochondrion on chloroplast surface was analyzed based on the time-lapse images of the Mt-RFP protoplast expressing the pGWT35S-Lifeact-Citrine gene. We characterized F-actin (green), mitochondrion (red), and chloroplast (blue), and then determined their colocalization by merged color (white) for over 12 s.”

We revealed that mitochondrion associated with F-actin on the chloroplast by providing additional Supplementary Fig. 4 and inserted sentences about the interaction as follow:

(Page 13, Lines 180-182)

“When mitochondria and F-actin on chloroplast surface were observed, we found that mitochondria in close proximity to F-actin on chloroplast (Supplementary Fig. 4, Supplementary Table 1).”

Supplementary Fig. 4 Association of mitochondria and F-actin on chloroplasts. Representative images are acquired from time-lapse analysis of three protoplasts (No.1-3) expressing both the pGWT35S-Lifeact-Citrine and MT-RFP for visualizing F-actin (green) and mitochondrion (magenta). The interaction between mitochondria, F-actin, and chloroplasts (blue) are shown (white arrows). Scale bars: 5 μ m.

Comment 8. In addition, how are the wiggling mitochondria defined in Fig. 5? The quantitative data should be included in Fig 5.

Response 8

Fig. 5 shows the effect of cytochalasin B on F-actin structure. F-actin structures were mostly disrupted in 50 μ M or 500 μ M cytochalasin B-treated cells. We selected representative images of the wiggling in Fig. 5b, which shows similar pattern of the trajectory and scatter plot (speed - angle changes in Supplementary Fig. 5) to that in F-actin-disrupted cell (Fig. 4c). In addition, the MSD analysis defined mitochondrial movements quantitatively in F-actin-disrupted cells as the wiggling as described in Response 1. We inserted sentences about how we defined the wiggling in F-actin-disrupted cell and the quantitative data as follows:

(Pages 13-14, Lines 189-194)

“In 500 μM cytochalasin-treated protoplasts, the mitochondrial movement was confirmed to be the wiggling by analyzing the tracking and speed - angle changes of mitochondrial movement (Supplementary Fig. 5), which shows similar pattern to the wiggling in Figs. 2-4. About 81% of mitochondria revealed the wiggling on chloroplast in 500 μM cytochalasin-treated protoplasts (Supplementary Fig. 6). These results confirm that mitochondrial wiggling on chloroplasts occurs independently of F-actin.”

Supplementary Fig. 5 Characterization of mitochondrial movements in cytochalasin-treated protoplasts expressing Lifeact-Citrine gene. **a, b** Representative trajectories (**a**) and scatter plot of speed and angle changes (**b**) of three movements of mitochondria obtained from three protoplasts treated with 500 μM cytochalasin.

Supplementary Fig. 6 Number of wiggling mitochondria in F-actin-disrupted protoplasts. Ratio of number of wiggling mitochondria with chloroplast to total number of mitochondria in cytochalasin-treated

protoplasts expressing pGWT35S-Lifeact-Citrine gene. * $P < 0.01$ (Student's t test).

Comment 9. Line 194-196, It is hard to understand the meaning. Please rewrite it.

Response 9

We considered the reviewer's opinion and decided to remove the sentence.

Comment 10. Line 200-202, The data is not meaningful because the range of data is less than the resolution of confocal microscopy.

Response 10

Thanks to point out our mistakes. We corrected these values as follows:

(Page 16, Lines 215-218)

“The mean distance per 5-s interval of mitochondria was $0.61 \pm 0.35 \mu\text{m}$, which was approximately two-fold greater than that per 250-ms interval ($0.35 \pm 0.14 \mu\text{m}$; Fig. 6b), suggesting that mitochondria can migrate in the intracellular space via wiggling.”

(Page 18, Lines 235-237)

“In the fixed cells, no significant difference was observed between the mean migration distance at the 250-ms ($0.33 \pm 0.08 \mu\text{m}$) and 5-s time intervals ($0.30 \pm 0.13 \mu\text{m}$) (Fig. 6c).”

Minor points:

Comment 11. Please check the positions of $>5 \mu\text{m}$ and $<5 \mu\text{m}$ in Fig 3b

Response 11

We thank the reviewer for pointing out our mistakes. We corrected the positions (Distance $<5 \mu\text{m}$ and $5 \mu\text{m} < \text{Distance}$) in Fig 3b.

Comment 12. In Supplementary Fig. 6, Constant association (CA) 0 Continuous association.

Response 12

We corrected the “constant” to “**continuous**” in Supplementary Fig.6 (Supplementary Fig.9 in revised manuscript). We changed “Constant association” to “Continuous association (CA)” in the other Figures.

Response to Reviewer #3

We sincerely appreciate Reviewer #3 for giving us insightful comments and suggestions, which have significantly improved our manuscript. We have carefully addressed to the reviewer comments as follows:

Reviewer #3 (Remarks to the Author):

This paper describes the different types of motility that are exhibited by mitochondria, changes in this behavior when in the vicinity of a chloroplast and the effect of actin and microtubule depolymerisation on motility of mitochondria.

Comment 1. When I read the manuscript I was surprised that many relevant papers that (in part) come to similar conclusions as in this paper were not cited. For example, the work of the Mathur group is completely missing (see e.g. Barton et al., JCS 2018), no literature is cited that covers the specific organisation and function of the peri-plastidal actin cytoskeleton (for review see e.g. Wada, Plant Sci. 2013) and Akkerman et al., Plant Cell Physiol 2011.

Response 1

Thank you very much for pointing out insufficiency of the references, which are related to our study. We agreed with the reviewer and added all the recommended references to Introduction or Discussion in the present manuscript. We added Barton et al, 2018¹⁰, on Page 3, Line 45. In addition, the wiggling in the present study is induced by the association with chloroplasts, which is a different mechanism from wiggling or F-actin-dependent system described in Akkerman et al, 2011²⁸ and Wada et al, 2013²⁹. We referred to these references in Discussion as follow:

(Page 30, Lines 417-419)

“Given that wiggling of mitochondria is independent on F-actin, regulation of mitochondria wiggling is different from wiggling of Golgi body depend on fine F-actin²⁸ or short-actin polymerization on chloroplast surface²⁹.”

Comment 2. There are significant overlaps between the work and conclusions in the current manuscript and the literature that is not cited.

Response 2

We thought that the significant overlaps, which the reviewer pointed out, were relationship between F-actin and mitochondrial movements and interaction between chloroplasts and mitochondria. We sincerely referred to the related papers in Introduction and Discussion (Pages 3-4, Lines 41-63; References No. 4 - 16 please see in References). We also referred to wiggling (Page 4, Lines 54-55) as follows:

“During treatment with F-actin-disrupting drugs, F-actin-independent wiggling of mitochondria was observed^{12,13}.”

In accordance with the reviewer’s comments, our results partially overlapped with the current reports

from other groups. However, our main finding that the wiggling is induced by the association with chloroplasts is different from the previous reports and an intriguing mechanism guided by trajectory analysis of mitochondrial movement in leaf mesophyll cells (migrate distance, speed, angle changes). We also performed MSD analysis of mitochondrial movements and concluded that directed- and diffusive motion with high value of the diffusion coefficient is a feature of the wiggling. We inserted the additional sentences, Figures, and Supplementary Table 4 about MSD analysis in Results as follows:

(Pages 24-25, Lines 303-320)

“MSD analysis of mitochondrial movement. To confirm characteristics of mitochondrial movements, we performed mean-squared displacement (MSD) analysis^{21,22,23,24} (Eq 1; see MSD calculation in Methods) on the trajectories of each mitochondrial movement in Figs. 2c, 4c, 6d, e, and 7c. The two-dimensional MSD plots revealed linear or parabolic shape, and were fitted to Eq 2 or Eq 3 (see MSD calculation in Methods, Supplementary Figs. 10-12, Supplementary Table 4). The fixed cell revealed linear shape defined as Brownian motion with low diffusion (D ; 0.00023)^{23,24,25} (Supplementary Fig. 10f). Mitochondrial movement with long-distance migration ($> 5 \mu\text{m}$), DMSO-treatment, oryzalin-treatment, NA, and PA exhibiting directed diffusion curve with steeper slope, meaning directed motion with high velocity (v : 0.406, 0.289, 0.263, 0.254, and 0.267, respectively) (Supplementary Figs. 10b, c, d and 11a, b). On the other hand, mitochondrial movement with short-distance migration ($< 5 \mu\text{m}$) and CA revealed both directed- and diffusive motion with slow slope (D : 0.11 and 0.029, respectively) (v : 0.0332 and 0.0436, respectively) (Supplementary Figs. 10a and 11c, Supplementary Table 4). Mitochondrial movement with cytochalasin-treatment also revealed both directed- and diffusive motion with slow slope, but low velocity (v : 0.0100) and low diffusion (D : 0.00010) (Supplementary Fig. 10e, Supplementary Table 4). These MSD results confirmed that the wiggling has diffusive motion with low velocity, and that directional movement has high velocity.”

(Pages 28-29, Lines 375-388)

“Furthermore, the MSD analysis clearly separated wiggling as a short-distance migration with diffusive movement on chloroplast, while directional movement as a long-distance migration with high speed independently of chloroplast (Supplementary Figs. 10 and 11, Supplementary Table 4). Both movements (wiggling and directional movement) are apparently different from Brownian motion. However, the MSD analysis of mitochondrial movement in cytochalasin-treated cell revealed low-diffusion coefficient and low velocity, meaning that mitochondria exhibited diffusion in short range in F-actin free condition (Supplementary Fig. 10e). It indicated that wiggling is independent of F-actin, but F-actin would contribute to extend migrate distance of the mitochondrial movement on chloroplast, because speed of mitochondrial movement in cytochalasin-treated cell was dramatically reduced, as compared to a mitochondrial movement with a short-distance migration (less than $5 \mu\text{m}$) and CA (Figs. 2, 4, and 7, Supplementary Figs. 2 and 3). While F-actin apparently has role in a long-

distance migration of mitochondria in cytosol as component of actomyosin system.”

Conclusively, we defined the wiggling and directional movement as follows:

(Page 30, Lines 408-415)

“Overall, the wiggling is defined as a mitochondrial movement possessing a short-distance migration with lower speed and high angle changes associated with high diffusion and low mean velocity derived from the MSD analysis, which is induced by interacting with chloroplast independently of F-actin. The short-distance below 5 μm is related to chloroplast size, which mitochondria associate with. The directional movement is defined as a mitochondrial movement possessing a long-distance migration with high speed and low angle changes associated with low diffusion and high mean velocity derived from the MSD analysis, which depends on F-actin.”

Supplementary Fig. 10 MSD analysis of mitochondrial movement. a-f Each plot is fitted to linear or curve models using the least-squares method with the Eq 2. **a, b** The mean squared displacement (MSD) of mitochondrial movements, which are separated to shorter (open circles) or longer (filled-magenta circles) than 5.0 μm distance. **c-e** MSD analysis for mitochondrial movement in DMSO-, oryzalin-, and cytochalasin-treated cells. **f** MSD analysis

for mitochondria movement in fixed cell.

Supplementary Fig. 11 MSD analysis of mitochondrial movement associating with or without chloroplast.

Each plot is fitted to curve models using the least-squares method with the Eq 2. **a-c** The mean squared displacement (MSD) analysis of three types of mitochondrial movements; no association (NA; **a**), partial association (PA; **b**), and continuous association (CA; **c**) with chloroplasts.

Supplementary Table 4

Parameter of MSD analysis and character of mitochondrial movements

	D ($\mu\text{m}^2/\text{s}$)	v ($\mu\text{m}/\text{s}$)	Pattern
MD < 5 μm	0.11	0.0332	Dire + Diff
5 μm < MD	-	0.406	Dire
DMSO	-	0.263	Dire
Oryzalin	-	0.289	Dire
• MD < 5 μm (Oryzalin)	0.038	0.134	Dire + Diff
• 5 μm < MD (Oryzalin)	-	0.475	Dire
Cytochalasin	0.00010	0.0100	Dire + Diff
NA	0.00075	0.254	Dire + Diff
PA	-	0.267	Dire
CA	0.029	0.0436	Dire + Diff
Fixed	0.00023	-	Brownian

D : Diffusion coefficient, v : mean velocity, MD: migrate distance, NA: No association with chloroplast, PA: partial association with chloroplast, CA: continuous association with chloroplast, Dire: Direct, and Diff: Diffusion.

Comment 3. Besides this problem, there are multiple issues, some of which are mentioned below: Why were protoplasts taken as a modelsystem? The authors repeat the experiments in intact leaf mesophyll cells, but the main results are obtained in protoplasts. It is uncertain how the absence of a cell wall or the removal from a tissue context affect the intracellular organisation or motility. Response 3

We agreed with the reviewer that intracellular organization or motility in the protoplast is different from that in the intact leaf cell. In fact, mitochondrial activity was slightly reduced in the protoplast, as compared to that in the intact leaf cell (Supplementary Fig. 13). However, we thought that the mesophyll protoplast has many advantages in CLSM analysis, as compared to the intact leaf mesophyll cell. Firstly, we focused on a single leaf mesophyll cell for preventing contamination of images of chloroplasts (small and poorly developed) and mitochondria from leaf epidermis cells, which are in the immediate upper layer of the mesophyll cells. Distance between bottom of a leaf epidermis cell and top of first layer of a leaf palisade mesophyll cell is too narrow to separate these cell border. Therefore, we thought that the mesophyll protoplast is suitable for CLSM analysis to obtain clear images by easily adjusting focus on mitochondria and chloroplasts. As shown in this study, the mesophyll protoplast is easy to examine the pharmacological

(e.g. cytochalasin, oryzalin) effects of mitochondrial movements. In addition, protoplast can be easily transformed by the appropriate plasmid vectors using PEG method⁴³ (Figs. 4 and 5).

However, we also thought that the mesophyll protoplast has also disadvantages, as the reviewer pointed out. Therefore, we performed additional experiments using the intact leaf mesophyll cells to confirm the wiggling induced by association with chloroplasts (Supplementary Figs. 1,7,8, and 13) and could obtain the similar results to that from the mesophyll protoplast. We discussed about these topics as follows:

(Pages 25-26, Lines 334-345)

“In addition, we applied our methods for evaluating mitochondrial movement to intact leaf mesophyll cells and obtained similar results (Supplementary Figs. 1, 7, 8, and 13, Supplementary Movie 8). Our findings confirm the notion that mitochondrial wiggling generally occurs in leaf mesophyll cells. Frequency analysis of the mean speeds of mitochondria revealed that the high-speed fraction was slightly larger in leaf mesophyll cells (Supplementary Fig. 13a) than that in protoplasts (Supplementary Fig. 3a, DMSO). Both the mean and maximum speeds of mitochondrial movement were higher in leaf mesophyll cells than that in protoplasts (Supplementary Fig. 13b, c), likely due to differences in cell shape or culture conditions between protoplasts and intact leaf cells. Since mitochondria in both cell types are active, we mainly used leaf mesophyll protoplasts to obtain clear image with avoiding contamination of mitochondrial images from leaf epidermis cells. Leaf mesophyll protoplasts is useful material for fluorescence imaging in only mesophyll cell.”

Comment 4. I presume that single z-plane time series were collected. How do the authors consider mitochondria that move in or out of the focal plane during their analyses?

Response 4

We configured each parameter in the CLSM for tracking most mitochondria within 30 s at high-speed rate (250 ms/frame) and slightly expanded a pinhole size from 1 airy unit to cover mitochondria in deep area. As a result, we could mostly track the mitochondria, however a few mitochondria still disappeared from focus plane to go behind chloroplasts as the reviewer’s comment. In that case, we could not trace the mitochondria for 30 s successively and removed it from data set. We added the detail information in Methods as follows:

(Page 34, lines 483-485)

“Time-lapse serial images of mitochondrial movement were obtained every 250 ms for 30 s with increasing size of a pinhole to 100 μm x 100 μm from 1 airy unit for tracking mitochondria in deep area.”

Comment 5. The conclusion that microtubules do not affect the movement of mitochondria is premature. Since microtubules are likely to have a cortical localisation, only the motility in this

plane should be considered. To link motility to microtubules, co-localisation of mitochondria and microtubules would be essential.

Response 5

In the previous-version manuscript, we concluded that microtubule was not involved in both directional movement and wiggling, because both the movements were kept in oryzalin-treated cell similar to that in control cells. However, we performed some additional experiments and reconsidered the conclusions in this revised manuscript, according to the reviewer's comment. Based on the additional results, we concluded that microtubule would slightly give effect on mitochondria directional movement. We would like to explain how microtubule contributed to mitochondrial movements as follows. Firstly, the scatter plot in oryzalin-treated cell revealed reduction of the spots of high speed - low angle change (Fig. 4c). Frequency of speed distribution of mitochondrial movements revealed reduction in high-speed area at more than $0.5 \mu\text{ms}^{-1}$ in oryzalin-treated cell as compared to that in DMSO (Supplementary Fig.3). Considering the reports from other groups, which mention about effect of the microtubule on mitochondrial movement (Gestel et al, 2002¹¹; Zheng et al, 2009¹²; Hamada et al, 2012²⁷), we concluded that microtubule would slightly give effect on mitochondria directional movement. The revised explanation was added in Results and Discussion as follows:

(Page 11, Lines 158-161)

“However, plots of speed (Fig. 4c) and speed frequency (Supplementary Fig. 3) at more than $0.5 \mu\text{ms}^{-1}$ were slightly decreased in oryzalin-treated cells. Therefore, both directional movement and wiggling occurred independently of microtubules, but microtubule may have effect on mitochondria directional movement.”

(Pages 26-27, Lines 352-360)

“However, treatment with oryzalin, a microtubule-disrupting drug, did not inhibit mitochondrial wiggling, whereas directional movement seemed to be slightly reduced in *A. thaliana* mesophyll protoplasts (Fig. 4a, c, Supplementary Fig. 3, Supplementary Movie 2), suggesting that this wiggling occurs independently, but the directional movement would be slightly affected by microtubules and related motors, such as kinesins. The result would be related to microtubule function affecting actin filament organization leading to affecting mitochondria velocity and trajectory in directional movement^{11,12}, or to the event that mitochondria trapped at F-actin - microtubule junction²⁷.”

On the other hand, we concluded that microtubule doesn't contribute to the wiggling, because the MSD of the mitochondrial movement in oryzalin-treated cell divided by the migrate distance (less or more than $5 \mu\text{m}$) was similar to that in control cell. We inserted the sentences about MSD analysis about microtubule and Fig. 12 as follows:

(Page 27, Lines 360-364)

“Our MSD analysis showed that mitochondrial movement with a short-distance migration ($< 5 \mu\text{m}$) in oryzalin-treated cell maintained both directed- and diffusive motion with slow slope ($D: 0.038$, $v: 0.134$) (Supplementary Fig. 12, Supplementary Table 4), which is characteristic of the mitochondrial wiggling. Thus, it appears that microtubules are not a driving force for the mitochondrial wiggling.”

Supplementary Fig. 12 MSD analysis of mitochondrial movement in oryzalin-treated cell. Each plot is fitted to curve models using the least-squares method with the Eq 2. The mean squared displacement (MSD) of mitochondrial movement in oryzalin-treated cells, which are separated to shorter (a) or longer (b) than 5.0 μm -migrate distance.

Comment 6. How do the authors define association with a chloroplast? An attachment is not shown, but it depends on the definition how this is interpreted.

Response 6

We thank the reviewer for providing such insightful comments. We defined interaction between mitochondrion and chloroplast based on the experimental results obtained from the following experiments in the present study. Firstly, we separated the mitochondrial movements to the wiggling and directional movement based on trajectory analysis of mitochondrial movements taken by CLSM at high-speed rate. Then, we defined association of mitochondria with chloroplast by measuring time when a mitochondrion associates with a chloroplast from the trajectory analysis described in Methods as follows:

(Page 36, Lines 511-513):

“To classify the dependence of mitochondrial movement on the association with chloroplasts, we analyzed the mitochondrial trajectories in the serial images. We classified all mitochondria into no, partial (more than 3 s), and continuous (within 30 s) association with chloroplasts.”

We showed that a mitochondrial movement in continuous association with chloroplast (CA) was the wiggling. In addition, we performed the MSD analysis of mitochondrial movements (Pages 24-25, Lines 303-320; Page 27, Lines 360-364; Page 28, Lines 375-388; and Page 30, Lines 408-415, see Response 1) for separating the wiggling associating with chloroplast from other type of mitochondrial movements. In

the present study, we gained the evidence of the direct interaction between mitochondria and chloroplast by performing biochemical co-isolation assay (Supplementary Fig. 14) and physiological analysis by measuring stable distance between mitochondria and chloroplast during time-lapse analysis (Supplementary Fig. 15; Supplementary Movie. 9). We inserted the corresponding sentences and put Figures below:

(biochemical co-isolation assay, Page 29, Lines 398-401)

“Moreover, western blot analysis of organellar proteins (mitochondria-localized MTS-Citrine and chloroplast-resident RuBisCO activase) showed that mitochondria were co-isolated with chloroplasts in the cytochalasin-treated leaves (Supplementary Fig. 14), indicating their interaction without F-actin.”

Supplementary Fig. 14 Co-isolation analysis of mitochondria and chloroplasts. a, b Western blot analysis of the isolated-chloroplast fraction (left) or total-leaf extract (right) using antibodies against GFP (a) or RuBisCO activase (RA) (b). MTS-Citrine at 28kD (a) and RA at 45kD (b) (arrow heads) are detected in both CP fraction and total leaf extraction from 0 μM, 50 μM and 500 μM cytochalasin-treated cells. c, d The images (c, d) show

the equal loading of western blotting proteins (a, b) after transferred onto the PVDF membrane and stained with the Ponceau S solution.

(Physiological analysis, Pages 29-30, Lines 401-405)

“We further examined the interaction of mitochondrion with chloroplast by measuring a distance between them in mobile chloroplasts during time-lapse analysis. The result showed that the distances were kept stably under half width of chloroplast diameter even though chloroplast moved vigorously, meaning that the interaction tightly occurred (Supplementary Fig. 15, Supplementary Movie 9).”

Supplementary Fig. 15 Interaction between mitochondrion and chloroplast in F-actin-disrupted protoplasts.

a Representative image of trajectories of mitochondria (magenta) and chloroplast (blue) in a cytochalasin-treated protoplast, which expresses pGWT35S-Lifeact-Citrine gene (green). Time-lapse images were acquired for 30 s at 1s intervals. Trajectory within 30 s of centroid of chloroplast (C) and mitochondrion (M) are shown as red color. Scale bar; 5 μm. **b** Distance between centroid of mitochondria and chloroplasts at each time points in 30 s are shown as lines. Each three set of chloroplast and mitochondrion (1-3) from three protoplasts (No. 1-3) is shown as different colors and lines (No. 1-1 to No. 3-3).

Comment 7. Since it is likely that these protoplasts have large central vacuoles, there may be a thin layer of cytoplasm that surrounds plastids. Mitochondria trapped in this cytoplasm may not

necessarily associated with a chloroplast.

Response 7

We observed not only wiggling but also directional movement on and near chloroplasts (e.g., see S1 Video). Given that mitochondria move freely on and near chloroplasts, it is considered that cytoplasmic thin layer around chloroplast does not mediate the association between mitochondria and chloroplasts. Therefore, we believed that there would be undefined factors directly connecting chloroplast with mitochondrion .

Comment 8. Besides this problem, the mitochondria in the vicinity of a chloroplast show a similar wiggling behavior as the mitochondria that are not associated (see e.g. movie 4).

Response 8

We checked the Supplementary Movie 4 again to doublecheck the wiggling behavior. Most of the mitochondria revealed the association with chloroplasts as marked with white asterisks (*) in the attached image below. As the reviewer pointed out, however, a few mitochondria in cytosol showed Brownian motion similar to that in fixed cells. To clear this technical issue, we removed those mitochondria from wiggling mitochondria in statistical analysis.

Supplementary Movie 4. Wiggling of mitochondria associated with a chloroplast in a CB-treated cell. Wiggling of mitochondria (arrow) associated with a chloroplast around the central area on the chloroplast at 0 s. White asterisks mean mitochondrion associated with chloroplast at the periphery.

We also performed statistical analysis about the number of mitochondria associated with chloroplasts in F-actin-disrupted cell, resulting that most of the mitochondria interacted with chloroplasts (Supplementary Fig. 6). Thus, to clear these points, we inserted the sentences about the interaction as follows:

(Page 29, Lines 393-395)

“Moreover, disrupting F-actin with cytochalasin increased the number of wiggling mitochondria

beside chloroplast (Figs. 4 and 5, Supplementary Figs. 2, 3, and 6, Supplementary Table 5, Supplementary Movies 3-5).”

Supplementary Table 5

Number of mitochondrion associating with or without chloroplast in 500 μ M CB-treated protoplasts

No.	With	Without	Total No.
Protoplast (n)	chloroplast (%) (SD)	chloroplast (%) (SD)	Mitochondrion (n)
10	92.8 (5.1)	8.1 (3.5)	726

Parenthesis: (SD; standard deviation), student’s t-test (With chloroplast vs Without chloroplast): $1.341E^{-19}$

Comment 9. In the absence of actin, something needs to generate a force for directional movement. That might be a flux in the cytoplasm, perhaps originating from membrane transport from plastid to cytoplasm and vice versa.

Response 9

The reviewer’s idea that membrane transport from plastid to cytoplasm tethering mitochondria is attractive and we added that discussion into Discussion as follows:

(Pages 31-32, Lines 437-441)

“Taken together, the two types of mitochondrial movement, wiggling and directional movement, would be related to energy supply and metabolic pathway among cellular compartments under the influence of cytoskeleton, cytoplasmic streaming, membrane transport, and undefined tethering factors in plant cells. The wiggling induced by association with chloroplast would have a significant meaning to affect chloroplast and cellular function.”

Comment 10. I would love to see decent statistics on the performed analyses.

Response 10

We followed to the reviewer’s suggestion and performed additional statistical analyses. We counted the number of the wiggling mitochondria in cytochalasin-treated protoplasts expressing Lifeact-Citrine (Supplementary Fig. 6), performed MSD analysis of mitochondrial movements (Supplementary Figs. 10, 11, and 12), measured time-dependent distance between a mitochondrion and a chloroplast (Supplementary Fig. 15), added Supplementary Tables 2 and 3 about the number of mitochondria in NA, PA, and CA groups, and added the statistical analysis of mitochondria associating with chloroplast in cytochalasin-treated cells (Supplementary Table. 5)

Reviewers' comments:

Reviewer #1 (Remarks to the Author):

The quality of the analysis of the mitochondrial motion has significantly improved. However I feel that the data is richer than the analysis and the authors should, in future, consider collaborations that could enhance the mathematical and statistical analysis. I still have a few concerns however.

1. Some parts of the analysis are still confusing. On lines 128-133 the authors claim that mitochondria that inhabit the grey region move less than 5 microns while those that inhabit the blue region move more than 5 microns. However this is contradicted by Fig 2c where the pink dots representing motion greater than 5 microns are found in both the grey and the blue regions. There may be a few more pink dots in the blue region than the grey but (i) this is not claimed and (ii) no numbers or statistical tests have been performed to suggest that the two populations are statistically different. This is even more pronounced in the leaf mesophyll data in Supp Fig 1. While the distinction between the two forms of movement are apparent in Fig 2c, it appears that mitochondria move long distances by "wiggling" too. However this is not noted or discussed.

2. Please correct the typos in the MSD analysis section. You have written liner for linear, sloop for slope.

3. I am confused about the results in Supp Fig 10. What is the difference between panel (a) and panel (c)? Does the data in panel (c) include all mitochondria, irrespective of the distance travelled? If that is so shouldn't we expect that it would be the same as the MSD analysis of panel (a) and (b) together? Or is DMSO making a difference to mitochondrial mobility? Again in Supp. Table 4 the DMSO treated mitochondria show only directed motion. There is similarly a difference between Fig 4c and Fig 2c.

4. There is no data on the goodness of fit (i.e confidence intervals or p-values) in the MSD analysis fits in Supp Table 4. So we cannot judge whether, for example, the velocity of 0.046 microns/s is significant or should be treated as effectively zero.

5. The conclusion that "the wiggling has diffusive motion with low velocity, and that directional movement has high velocity." in lines 319 and 320 could be enhanced. In this section you currently only report the numbers in Supp table 4. However the important point here are the comparisons between the numbers. Thus you could point out that CA mitochondria show negligible directed motion compared with NA mitochondria. Thus all the previous conclusions can be recapitulated through this analysis. There are some new points that you could also make. For example, the diffusion constant is greatly reduced by cytochalasin, even when comparing with the <5micron population, suggesting that mitochondrial wiggling may be related with actin. Thus wiggling does not represent thermal diffusion but random motion due to cytoskeleton related activity (possibly powered by the mitochondria?). I suggest that instead of merely reporting the numbers you report on the comparisons between treatments, concentrating on the fit parameters that are statistically significant

Response to Reviewer #1

We sincerely express our appreciation to the Reviewer #1 for carefully reading our revised manuscript and providing us critical comments and advices to further improve our manuscript. We have addressed the reviewer's comment point by point as follows:

Reviewer #1 (Remarks to the Author):

The quality of the analysis of the mitochondrial motion has significantly improved. However I feel that the data is richer than the analysis and the authors should, in future, consider collaborations that could enhance the mathematical and statistical analysis.

I still have a few concerns however.

Comment 1. Some parts of the analysis are still confusing. On lines 128-133 the authors claim that mitochondria that inhabit the grey region move less than 5 microns while those that inhabit the blue region move more than 5 microns. However this is contradicted by Fig 2c where the pink dots representing motion greater than 5 microns are found in both the grey and the blue regions. There may be a few more pink dots in the blue region than the grey but (i) this is not claimed and (ii) no numbers or statistical tests have been performed to suggest that the two populations are statistically different. This is even more pronounced in the leaf mesophyll data in Supp Fig 1. While the distinction between the two forms of movement are apparent in Fig 2c, it appears that mitochondria move long distances by "wiggling" too. However this is not noted or discussed.

Response 1

Thank you for pointing out our insufficient interpretation of distribution of different type (short and long distance migration, black and magenta) of mitochondria in two area (grey and blue). To address the reviewer's comments, we have additionally performed statistical tests for the two different type of the mitochondria belong to short- and long-distance migration. We inserted additional Supplementary Fig. 2 and sentence as follows;

(Pages 9-10, Lines 135-138)

[These different types of mitochondria, which were separated depend on migrate distance, had statistically different mean speeds and angle changes (Supplementary Fig. 2), however a long-distance migration of mitochondria had partially the plot of low speed and high-angle changes.]

Supplementary Fig. 2 Comparison of mean speeds and angle change of mitochondria between short- or long-distance migration. a,c Mean speed of mitochondria from both mitochondria in short ($D < 5\mu\text{m}$)- and long ($5\mu\text{m} < D$)- distance migration in protoplasts (**a**) and leaf mesophyll cells (**c**). **b,d** Mean angle change (Δ Angle (θ)) of mitochondrial movement from both mitochondria in short ($D < 5\mu\text{m}$)- and long ($5\mu\text{m} < D$)- distance migration in protoplast (**b**) and leaf mesophyll cell (**d**). * $P < 0.01$ (Student's- t test; 3.39E-06 in (**a**), 9.01E-08 in (**b**), 7.39E-20 in (**c**), and 2.26E-22 in (**d**)).

As the reviewer mentioned, the mitochondrion migrates more than $5\mu\text{m}$ has also plots belong to low speed and high-angle changes and vice versa. Representative three mitochondria plots in Fig.3 (b) also revealed a similar trend. During mitochondrial movement within 30 s, both type of mitochondria have lower number of extra plots. We also inserted a word [**largely**] as follows;

(Page 10, Lines 139 and 140)

[Taken together, mitochondria migrating less than $5\mu\text{m}$ in 30 s moved largely at low speed with high-angle change rates defined as wiggling, and mitochondria migrating more than $5\mu\text{m}$ in 30 s moved largely at high speed with low-angle change rates defined as directional movement.]

Comment 2. Please correct the typos in the MSD analysis section. You have written liner for linear, sloop for slope.

Response 2

We are sorry for our mistakes. In the revised version, we have corrected the miss typos at Page 24, Line 311 and 313, and at Page 39, Line 567 to [**linear**], and Page 24, Line 316 to [**slope**].

Comment 3. I am confused about the results in Supp Fig 10. What is the difference between panel (a) and panel (c)? Does the data in panel (c) include all mitochondria, irrespective of the distance travelled? If that is so shouldn't we expect that it would be the same as the MSD analysis of panel (a) and (b) together? Or is DMSO making a difference to mitochondrial mobility? Again in Supp. Table 4 the DMSO treated mitochondria show only directed motion. There is similarly a difference between Fig 4c and Fig 2c.

Response 3

We are sorry for this confusing description. As the reviewer's comment, the Fig. 10 (c) is the MSD about all mitochondrial movement in DMSO treated-protoplasts examined as control for oryzalin- or cytochalasin-treated protoplast. The Fig. 10 (a) and (b) are the MSD about mitochondrial movement, which were separated depend on the migrate distance in non-treated protoplast. We used low concentration of DMSO, however we could not ignore the effect of DMSO on the mitochondrial movement. In addition, number of mitochondria traveling at long distance are less than that at short distance, therefore the MSD of the Fig. 10 (c) would be theoretically less than the MSD about simple sum of (a) and (b). The MSD curve of the DMSO (c) shows only directed motion, however the Fig. 4 (c) and Fig. 2 (c) contain the wiggling. We think that diffusive parameter would be hidden behind direct motion of mitochondrial movement, which mostly increased speed in time-dependent manner. We examined the MSD analysis about mitochondrial movement separated to the short- and long-migrate distance.

Comment 4. There is no data on the goodness of fit (i.e confidence intervals or p-values) in the MSD analysis fits in Supp Table 4. So we cannot judge whether, for example, the velocity of 0.046 microns/s is significant or should be treated as effectively zero.

Response 4

Based on the reviewer's comment, we additionally calculated the chi-squared value (χ^2) to test the goodness of fit about fitting curve of the MSD analysis and calculated p-value. The results showed that the MSD analysis fits are reliable. To explain this point, we inserted an additional sentence to the Methods with Supplementary Table 4 including each value as follows:

(Page 40, Lines 577-578)

[The chi-squared value, χ^2 was used to test the goodness of fit of the MSD analysis fits conditioned on a null hypothesis (Supplementary Table 4).]

	D ($\mu\text{m}^2/\text{s}$)	v ($\mu\text{m}/\text{s}$)	Pattern	χ^2	P – value
MD < 5 μm	0.11	0.0332	Dire + Diff	1.00	0.317
5 μm < MD	-	0.406	Dire	6.69E-05	0.993
DMSO	-	0.263	Dire	1.00	0.317
Oryzalin	-	0.289	Dire	0.419	0.517
MD < 5 μm (Oryzalin)	0.038	0.134	Dire + Diff	1.00	0.317
5 μm < MD (Oryzalin)	-	0.475	Dire	0.931	0.334
Cytochalasin	0.00010	0.0100	Dire + Diff	1.00	0.317
NA	0.00075	0.254	Dire + Diff	1.00	0.317
PA	-	0.267	Dire	1.00	0.317
CA	0.029	0.0436	Dire + Diff	0.587	0.444
Fixed	0.00023	-	Brownian	1.00	0.317

Supplementary Table 4. Parameter of MSD analysis and character of mitochondrial movements. D : Diffusion coefficient, v : mean velocity, MD: migrate distance, NA: No association with chloroplast, PA: partial association with chloroplast, CA: continuous association with chloroplast, Dire: Direct, and Diff: Diffusion. The chi-squared value, χ^2 and p-value to test the goodness of fit about the MSD analysis fits are shown.

Comment 5. The conclusion that "the wiggling has diffusive motion with low velocity,

and that directional movement has high velocity." in lines 319 and 320 could be enhanced. In this section you currently only report the numbers in Supp table 4. However the important point here are the comparisons between the numbers. Thus you could point out that CA mitochondria show negligible directed motion compared with NA mitochondria. Thus all the previous conclusions can be recapitulated through this analysis.

There are some new points that you could also make. For example, the diffusion constant is greatly reduced by cytochalasin, even when comparing with the <5micron population, suggesting that mitochondrial wiggling may be related with actin. Thus wiggling does not represent thermal diffusion but random motion due to cytoskeleton related activity (possibly powered by the mitochondria?).

I suggest that instead of merely reporting the numbers you report on the comparisons between treatments, concentrating on the fit parameters that are statistically significant.

Response 5

Thank you for providing significant suggestion, we have considered the MSD results and inserted the sentence as follows :

(Page 24, Lines 321-322)

[This means that CA shows negligible directed motion compared with NA, and is well suited for wiggling.]

We have also inserted the sentence about MSD results based on the reviewer's suggestion as follows;

(Pages 24-25, Lines 324-326)

[These MSD results conclude that the wiggling has diffusive motion with low velocity, and that directional movement has high velocity, supporting the previous conclusion in this study.]

About reduction of the diffusion constant in the MSD analysis for cytochalasin, we discussed at Page 28, lines 384-393 in previous revised manuscript. Here, we have again modified the sentence following to the reviewer's suggestion in Abstract, and Discussion as follows;

(Page 2, Lines 28-33 in Abstract)

[The MSD analysis could separate these two movements. Directional movement was dependent on filamentous actin (F-actin), whereas mitochondrial wiggling was not, but slightly influenced by F-actin. In mesophyll cells, mitochondria could migrate by wiggling, and most of these mitochondria associated with chloroplasts. Thus,

mitochondria migrate via F-actin-independent wiggling under the influence of F-actin during their association with chloroplasts in *Arabidopsis*.]

(Pages 28-29, Lines 386-393 in Discussion)

[However, the MSD analysis of mitochondrial movement in cytochalasin-treated cell revealed low-diffusion coefficient and low velocity, even when comparing with a mitochondrial movement with a short-distance migration (less than 5 μm) and CA (Figs. 2, 4, and 7, Supplementary Figs. 3, 4 and 10), suggesting that the wiggling may be influenced by F-actin. It means that F-actin would contribute to extend migrate distance of the mitochondrial movement on chloroplast. Therefore, the wiggling does not represent thermal diffusion, but random motion including cytoskeleton related activity.]

REVIEWERS' COMMENTS:

Reviewer #1 (Remarks to the Author):

While many of the concerns I raised previously have been addressed, there are still minor concerns regarding the changes that the authors have made that should be addressed. I list them below.

1. In their rebuttal letter Response 1. the authors say that they inserted the following sentence: "These different types of mitochondria, which were separated depend on migrate distance, had statistically different mean speeds and angle changes (Supplementary Fig. 2), however a long-distance migration of mitochondria had partially the plot of low speed and high-angle changes.]"

Unfortunately the sentence is badly constructed. Please correct the language.

2. In Response 4, Supplementary Table 4 I am assuming that the null hypothesis was that the model (i.e. diffusion or diff plus drift) fitted the data, hence the p-values did not imply rejection of the null hypothesis. If this isn't true, the p-values are too high. It may be a good idea to clarify that here to prevent confusion in readers.

3. In Response 5 the authors have inserted: [This means that CA shows negligible directed motion compared with NA, and is well suited for wiggling.]

Unfortunately the phrase "well suited for wiggling" does not make sense. Please rewrite.

4. In Response 5. the authors have included the sentence: [These MSD results conclude that the wiggling has diffusive motion with low velocity,...].

The phrase "wiggling has diffusive motion" should be replaced by "wiggling appears to be similar to diffusive..."

Response to Reviewer #1

We sincerely express our appreciation to the Reviewer #1 for carefully reading our revised manuscript again and providing us critical and adequate comments to improve our manuscript. We have addressed the reviewer's comments point by point as follows:

REVIEWERS' COMMENTS:

Reviewer #1:

Remarks to the Author:

While many of the concerns I raised previously have been addressed, there are still minor concerns regarding the changes that the authors have made that should be addressed. I list them below.

Comment 1. In their rebuttal letter Response 1. the authors say that they inserted the following sentence: "These different types of mitochondria, which were separated depend on migrate distance, had statistically different mean speeds and angle changes (Supplementary Fig. 2), however a long-distance migration of mitochondria had partially the plot of low speed and high-angle changes." Unfortunately the sentence is badly constructed. Please correct the language.

Response: We have corrected and shorten the sentence as follows;

(Page 7, Lines 109-111)

“These different types of mitochondria, which were separated depending on migrate distance, had different mean speeds and angle changes (Supplementary Fig. 2).”

Comment 2. In Response 4, Supplementary Table 4 I am assuming that the null hypothesis was that the model (i.e. diffusion or diff plus drift) fitted the data, hence the p-values did not imply rejection of the null hypothesis. If this isn't true, the p-values are too high. It may be a good idea to clarify that here to prevent confusion in readers.

Response: Thank you for your suggestion. The high p-values are high and we accepted the fitting model for the MSD analysis. Based on the comments, we revised the main text as follows:

(Page 30, Line 501-502)

“The chi-squared value χ^2 was used to test the goodness of fitting of the MSD analysis. The fitting to a null hypothesis indicates that the model fits the data. The high p-values does not mean rejecting the null hypotheses (Supplementary Table 4).”

Comment 3. In Response 5 the authors have inserted: [This means that CA shows negligible

directed motion compared with NA, and is well suited for wiggling.]

Unfortunately the phrase "well suited for wiggling" does not make sense. Please rewrite.

Response: We have rewritten the sentence as follows:

(Pages 15, Lines 239-240)

“This means that CA shows negligible directed motion compared with NA, and seems to be wiggling.”

Comment 4. In Response 5. the authors have included the sentence: [These MSD results conclude that the wiggling has diffusive motion with low velocity,...]. The phrase "wiggling has diffusive motion" should be replaced by "wiggling appears to be similar to diffusive..."

Response: We have replaced the sentence as follows:

(Pages 15, Lines 243-245)

“These MSD results conclude that the wiggling appears to be similar to diffusive motion with low velocity, and that directional movement has high velocity, supporting the previous conclusion in this study.”